Glacial Isostatic Adjustment modelling: historical perspectives, recent advances, and future directions

# Pippa L. Whitehouse[1]

[1]Department of Geography, Durham University, Durham, DH1 3LE, UK

*Correspondence to*: Pippa L. Whitehouse (pippa.whitehouse@durham.ac.uk)

**Abstract.** Glacial Isostatic Adjustment (GIA) describes the response of the solid Earth, the gravitational field, and the oceans to the growth and decay of the global ice sheets. A commonly-studied component of GIA is 'postglacial rebound', which specifically relates to uplift of the land surface following ice melt. GIA is a relatively rapid process, triggering 100 m-scale changes in sea level and solid Earth deformation over just a few tens of thousands of years. Indeed, the first-order effects of GIA could already be quantified several hundred years ago without reliance on precise measurement techniques and scientists

have been developing a unifying theory for the observations for over 200 years. Progress towards this goal required a number of significant breakthroughs to be made, including the recognition that ice sheets were once more extensive, the solid Earth changes shape over time, and gravity plays a central role in determining the pattern of sea-level change. This article describes the historical development of the field of GIA and provides an overview of the processes involved. Significant recent progress has been made as concepts associated with GIA have begun to be incorporated into parallel fields of research; these advances

are discussed, along with the role that GIA is likely to play in addressing outstanding research questions within the field of Earth system modelling.

## 1 Introduction

The response of the solid Earth to the collapse of northern hemisphere ice sheets following the Last Glacial Maximum (LGM, ~21,000 years ago) continues today at a rate so large (>10 mm/yr, e.g. Lidberg et al., 2010; Sella et al., 2007) that glacial

isostatic adjustment (GIA) is one of the few geophysical processes that can be readily observed on human timescales without recourse to sophisticated scientific measurement techniques. For this reason, the more easily-observed impacts of GIA, such as shoreline migration, played an important role in motivating the development of ideas associated with climate cycles, sea-level change, geodesy and isostasy during the 19th and early 20th century (Jamieson, 1865; Croll, 1875; Woodward, 1888; Nansen, 1921; Daly, 1925). The field has a long history of combining observations with theory, and the exchange of ideas

between scientists working in a suite of different disciplines has repeatedly resulted in important scientific breakthroughs.

The modern field of GIA addresses the classic geodynamics problem of determining the solid Earth response to surface load changes by ice and ocean water, whilst at the same time solving for the gravitationally-consistent redistribution of meltwater across the global ocean. Calculations are necessarily carried out on a global scale, and numerical models of GIA consider the

behaviour of three fundamental components of the Earth system: the solid Earth, the global ocean, and the ice sheets (see Figure 1). Inputs to a GIA model typically include a priori estimates for the history of global ice-sheet change and the rheology

of the solid Earth, with changes to the shape of the ocean and the solid Earth being determined by solving the sea-level equation (Farrell and Clark, 1976; see also Sect. 2.2). A wealth of data can be used to determine the details of GIA model inputs, e.g. geological evidence can provide information on past ice extent (Bentley et al., 2014), while modelling of mantle dynamics can be used to determine independent constraints on mantle viscosity (e.g. Mitrovica and Forte, 2004). However, the reason GIA is of interest across so many disciplines is that the problem can be turned around, and observations relating to past sea-level change or solid Earth deformation – the classical 'outputs' of a GIA model (Figure 1) – can be used to infer information relating to the 'inputs', namely ice-sheet history and Earth rheology (e.g. Lambeck et al., 1998; Peltier, 2004). As in all disciplines where data play a crucial role in determining model parameters, uncertainties and spatial/temporal gaps in the data leave room for non-uniqueness in the solutions invoked to explain the observations, but intellectual input from a diverse range of sources over the last 200 years has helped to steer the field towards robust explanations for the varied range of processes that are associated with GIA.

In this article the development of GIA modelling is traced from initial observations of rapid shoreline migration in 15[th] century Sweden through to sophisticated approaches that incorporate feedbacks between ice, ocean and solid Earth dynamics. The historical development of the field is described in Sect. 2.1 and the remainder of Sect. 2 provides an overview of the underlying theory, important results, and data sets used to constrain GIA modelling – more detailed reviews of the technical aspects of GIA modelling can be found elsewhere (e.g. Whitehouse, 2009; Steffen and Wu, 2011; Milne, 2015; Spada, 2017). Recent developments in the field are discussed in Sect. 3, and the article concludes with a discussion of unresolved questions that warrant future attention (Sect. 4). We begin by motivating this review with a brief summary of the fields that have been influenced by studies of GIA.

## 1.1 Applications of GIA

GIA plays a role in studies that span the fields of climate, cryosphere, geodesy, geodynamics, geomorphology, and natural hazards. Some of the fundamental scientific questions that require consideration of GIA include: (i) linking ice-sheet response to past climatic change; (ii) understanding the rheology of the interior of the Earth; (iii) determining present-day ice sheet mass balance and sea-level change; (iv) interpreting palaeo-sea-level records; (v) understanding ice-sheet dynamics; (vi) quantifying tectonic hazard; (vii) reconstructing palaeo-drainage systems; (viii) interpreting the gravity field and rotational state of the Earth; (ix) understanding coastal change and past migration routes; and (x) understanding the causes of volcanism. Comparison with data is a central component of GIA studies, and in many cases misfits between observations and predictions have led to new understanding of factors that had not previously been considered, such as feedbacks between GIA and ice dynamics or the influence of lateral variations in mantle rheology. The historical development of this multi-faceted subject is an interesting story.

## 2 Review of GIA modelling

### 2.1 Historical perspective

The people of Scandinavia must have been aware of the effects of GIA for many centuries. As an example, by 1491 AD the ancient port of Östhammer could no longer be reached by boat and the city had to be relocated closer to the sea (Ekman, 2009). However, it was not until the first half of the 18th century that rigorous measurements of relative sea-level change – i.e. change in local water depth – were initiated with the cutting of a series of 'mean sea level marks' into coastal rocks around Sweden (Ekman, 2009). Using historical documents to extend the record of sea-level change back to 1563 AD, Celsius (1743) carried

out the first calculations associated with GIA and determined that sea-level in the Gulf of Bothnia was falling at a rate of 1.4 cm/yr relative to the height of the land. He assumed that the change was due to a fall in sea level but at the time there was considerable disagreement, with others proposing that the cause was land uplift (see Ekman (2009) for a thorough review of the subject).

The question was partly resolved by considering evidence for relative sea-level change in different locations around the world. Playfair (1802) noted that past sea levels had been higher in such diverse locations as Scotland, the Baltic, and the Pacific, but lower in the Mediterranean and southern England. Ideas associated with the concept of an equipotential surface were yet to be put forward (e.g. Stokes, 1849), and therefore Playfair (1802) argued that since "the ocean… cannot rise in one place and fall in another" the differences must be associated with land level changes. Without any means to determine the timing of past

relative sea-level change in different locations this argument is flawed, but more robust evidence was provided by Lyell (1835), who used an ingenious variety of observations to determine that the *rate* of relative sea-level change across Sweden varied from place to place. Following a similar argument to Playfair (1802), Lyell (1835) concluded that his observations could only be explained by variations in the rate of land uplift, since (he assumed) sea-level fall would produce a spatially-uniform rate of change.


These early studies explored a number of explanations for the change in the height of the land, and the idea that an ice sheet could have depressed the land was first proposed by Jamieson (1865). He was familiar with the geomorphological evidence for past ice cover across Scotland, but made the important observation that whilst marine deposits could be found well above current sea level near the coast, they were not present in lower areas in the interior. This led him to propose that the weight of

the ice sheet must have depressed the coastal land below sea level, but that the presence of the ice prevented the interior from being flooded. This was the first suggestion that an ice sheet could deform the Earth, and whilst such ideas were relatively quickly taken up by field scientists (e.g. Chamberlin and Salisbury, 1885), they were not widely accepted by those who took a more theoretical approach, largely due to ongoing disagreement regarding the structure of the interior of the Earth and the concept of isostasy (e.g. Barrell, 1919). Crucially, the idea that the surface of the Earth could deform in response to a change

in surface load was neglected by those who first considered the effect of gravity on sea level.

An important contribution to the field came from Croll (1875), who proposed that there had been repeated glacial cycles and hence periodic changes in the distribution of mass throughout the Earth system. Based on his assumption that the ocean takes on a spherical form around the centre of mass of the Earth, he calculated the displacement of the centre of mass due to the presence of an ice cap, and found that the centre of mass of the system would be displaced towards the ice cap, thus providing an explanation for the observation that sea levels were higher during glacial times in Scotland. However, Croll's (1875) theory was flawed in two important ways: (i) he dismissed solid Earth deformation to be a local effect and did not link it to ice loading, and (ii) he did not appreciate that the redistribution of mass throughout the Earth system would alter the *shape* of the ocean surface. This second issue was addressed in detail by Woodward (1888) around a decade later.

Woodward's (1888) interest in the shape of the Earth's gravitational field was motivated by questions posed to him regarding the differing elevations of contemporaneous palaeo-shorelines of the former Lake Bonneville (Gilbert, 1885b), and the tilt of lake shorelines and glacial deposits associated with the past glaciation of North America (Chamberlin and Salisbury, 1885; Gilbert, 1885a). The authors of these studies were supporters of the hypothesis that surface load changes, in the form of water, ice or sediment, could deform the surface of the Earth, and they deduced that the solid Earth response to surface loading could provide an explanation for their observations (Gilbert, 1885b; Chamberlin and Salisbury, 1885). Indeed, Gilbert (1890) used palaeo-shoreline observations from Lake Bonneville to draw early conclusions on the rheological properties of the Earth. However, building on the ideas of Croll (1875), this group of scientists also wondered whether gravitational attraction, e.g. of an ice sheet, played a role in explaining their observations.

They turned to Woodward (1888) for the answer, and he carried out detailed calculations relating to the change in the shape of the geoid that would arise due to the redistribution of surface mass associated with the appearance/disappearance of an ice sheet. He used realistic estimates for the shape and size of an ice sheet, and took into account the self-attraction of the ocean and the different densities of ice and water. He also appreciated the need to conserve mass when transferring water between the ice sheets and the ocean, but addressed a simplified problem in which ice is transferred directly between the two polar regions (cf. Croll, 1875), i.e. his calculations assumed no net change in ocean volume. The key result of this work was a prediction of the perturbation to the height of the geoid at a series of radial distances from the centre of a growing ice sheet. Although he did not formally account for changes in ocean volume, the magnitude of the geoid perturbation in the near-field of the ice sheet led Woodward (1888) to hypothesize that as the ice sheet formed, water depths in the near-field would increase, despite a net decrease in ocean volume – a result that still surprises many people today!

The one shortcoming of Woodward's (1888) analysis was his decision to neglect the deformation of the Earth in response to surface loading. This led him to conclude that the volume of ice needed to explain the tilt of palaeo-shorelines in the Great Lakes region (Gilbert, 1885a) was unfeasibly large. If he had been able to include an estimate of Earth deformation he may

well have realized that the palaeo-shorelines could be explained by a combination of postglacial rebound and tilting of the geoid surface due to the attraction of the former ice sheet (e.g. Figure 2).

Important advances towards understanding the role of Earth deformation were made by Nansen (1921, p.288), who used the concepts of isostasy and mass balance to show that sea-level change could be explained by some combination of ice-sheet melt

and land deformation: "along great parts of the coasts of Fenno-Scandia, this rise of sea-level was more or less masked by the still faster upheaval of the land, and it was only during certain periods when the temperature was much raised and the melting of the ice caps much increased, that the rise of sea-level was sufficiently rapid to cause a pause in the negative shift of the shoreline so considerable that conspicuous marine terraces, beaches, or shorelines could be developed."

Nansen (1921) did not consider the gravitational effect of mass redistribution when discussing the causes of relative sea-level change, but he did understand that the Earth would continue to deform viscously for a prolonged period following mass redistribution, and hence that relative sea-level change would continue after the volume of the ocean stabilized. This reasoning led him to suggest that ongoing subsidence of the seafloor due to the past addition of meltwater to the ocean could explain observations of recent global sea-level fall (Daly, 1920) and that the growth and decay of peripheral bulges within the ocean

need to be accounted for when calculating the magnitude of sea-level change during a glacial cycle. This latter point is an early description of the 'ocean syphoning' effect (see Sect. 2.2.3 and Figure 2b). Daly (1925) further explored the implications of solid Earth deformation when seeking to interpret an impressive array of sea-level observations from around the world, making use of the idea that the Earth would respond both elastically and viscously in response to surface loading. He also highlighted the important, but little-known, work of Rudzki (1899), who took Woodward's (1888) geoid calculations and used them to re-

calculate the spatially-variable sea-level change that would result from the melting of a circular ice cap, but now accounting for the combined effects of (elastic) Earth deformation, the change in ocean volume, and the change in the shape of the gravitational field due to the redistribution of both ice and solid Earth mass.

By the 1920's a range of approaches had been used to estimate the magnitude of the sea-level lowstand at the peak of the last

glaciation (Daly, 1925). Nansen's (1921) estimates for global ice volume were based on calculations linking the magnitude of depression beneath the former ice sheets (evidenced by marine deposits, which are now located above present sea level) to the relative density of ice and upper mantle material, accounting for the fact that some mantle material would be laterally displaced to form peripheral bulges. Impressively, his estimate for the mean sea-level change associated with "the formation of the Pleistocene ice-caps" was 130 m, a value that is almost identical to contemporary estimates (Lambeck et al., 2014). Nansen's

(1921) calculations accounted for changes in the area of the ocean through time, and he identified several factors that necessitate an iterative approach to calculating changes in sea level. In particular, he noted that since ice loading deforms both continental and oceanic areas, the true change in water depth (which determines the deformation due to ocean loading) will be different to the value that would be determined by considering a constant ocean area and a non-deforming Earth. These

feedbacks between ice and ocean loading are a fundamental feature of the sea-level equation (Farrell and Clark, 1976), which forms the basis of most contemporary GIA models.

These early scientists made impressive use of the data available to them, and the final piece of the puzzle came with the ability to determine the timing of past environmental change (e.g. De Geer, 1912), which allowed the first estimate for the viscosity of the mantle to be determined (Haskell, 1935; ~$10^{21}$ Pa s for the upper mantle). Studies into the viscosity of Earth's mantle developed rapidly from the 1960's onwards (e.g. McConnell, 1968; O'Connell, 1971; Peltier, 1974; Cathles, 1975), and the stage was set for a global model of GIA that accounted for (i) ice-ocean mass conservation, (ii) viscoelastic deformation of the Earth, and (iii) gravitationally self-consistent perturbations to the shape of the geoid. The following section outlines the modelling approach that was developed in the 1970's to address these issues. The models developed during this period underpin all contemporary studies of GIA, and provide confirmation of many of the fundamental ideas proposed during the 19[th] and early 20[th] century.

## 2.2 Development of the sea-level equation

### 2.2.1 The original form of the sea-level equation

When modelling processes associated with GIA, water from a melting ice sheet is assumed to be instantaneously redistributed across the global ocean according to the shape of the geoid, where the geoid is the equipotential surface that defines the shape of the sea surface in the absence of dynamic forcing by atmospheric or oceanic circulation. The shape of the geoid depends on the distribution of mass throughout the Earth system. There are feedbacks to be considered because the change in the distribution of surface mass (e.g. the decrease in the mass of the ice sheet and the addition of mass to the ocean) must be taken into account when calculating the shape of the geoid as the meltwater is redistributed (Figure 2). However, the situation is more complicated than this because the shrinking of the ice sheet and the transferral of water to the ocean causes the solid Earth to deform, and this redistribution of mass *inside* the Earth further alters the shape of the geoid. Calculations to determine the change in sea level due to the melting of an ice sheet must therefore be carried out iteratively. The problem is further complicated by the fact that the deformation of the Earth reflects both contemporary and past surface mass change due to the viscoelastic properties of the mantle (Cathles, 1975).

There are two fundamental unknowns within GIA: the history of the global ice sheets and the rheology of the solid Earth (Figure 1). These are traditionally determined via an iterative approach, using a range of data (see Sect. 2.4). Once they are known, or once a first estimate has been determined, then the spatially-varying history of relative sea-level change can be uniquely determined by solving the sea-level equation, which defines the gravitationally self-consistent redistribution of meltwater across the ocean. The theoretical development of the sea-level equation is covered extensively elsewhere (e.g. Farrell and Clark, 1976; Mitrovica and Milne, 2003; Spada and Stocchi, 2006; Spada, 2017), and hence we briefly restate the main form of the equation here, starting with a definition of relative sea level:

$$S = N - U \tag{1}$$

Here, $S$ is relative sea level, or water depth; $N$ is absolute sea level, defined as the height of the sea surface above the centre of mass of the solid Earth, and $U$ is the height of the seafloor, again defined relative to the centre of mass of the solid Earth. From Eq. (1) it is clear that changes in relative sea level ($\Delta S$) arise due to a combination of changes to the height of the sea surface and the seafloor. Deformation of these two surfaces occurs in response to ice and ocean load changes, as calculated within the sea-level equation:

$$\Delta S(\theta, \psi, t) = \frac{\rho_i}{\gamma} G_S \otimes_i I + \frac{\rho_w}{\gamma} G_S \otimes_o \Delta S + C_{SL}(t) \tag{2}$$

$\Delta S(\theta,\psi,t)$ is the change in relative sea level (or, equivalently, water depth) at co-latitude $\theta$ and longitude $\psi$, between time $t$ and some reference time $t_0$, $I$ is the spatio-temporal evolution of global ice thickness change, $\rho_i$ and $\rho_w$ are ice and ocean water density, respectively, $\gamma$ is the acceleration due to gravity at Earth's surface, $G_S$ represents a Green's function that describes perturbations to the solid Earth displacement field and the gravitational potential due to surface loading, constructed by combining viscoelastic surface load Love numbers (Peltier, 1974; Spada and Stocchi, 2006), and $\otimes_i$ and $\otimes_o$ represent convolutions in space and time over the ice sheets and the ocean, respectively. Note the appearance of $\Delta S$ on both sides of Eq. (2), indicating that the sea-level equation is an integral equation and an iterative approach is required to solve it. The first two terms on the right-hand side of Eq. (2) are spatially-varying terms that describe the perturbation to sea level due to ice and ocean loading respectively, while $C_{SL}(t)$ is a time-dependent uniform shift in relative sea level that is invoked to satisfy conservation of mass:

$$C_{SL}(t) = -\frac{m_i(t)}{\rho_w A_0(t)} - \frac{\rho_i}{\gamma} \overline{G_S \otimes_i I} - \frac{\rho_w}{\gamma} \overline{G_S \otimes_o \Delta S} \tag{3}$$

The first term on the right-hand side of Eq. (3) is often referred to as the 'eustatic' term; it describes the spatially uniform sea-level change that takes place across ocean area $A_0(t)$ due to a change in ice mass of magnitude $m_i(t)$, in the absence of any solid Earth deformation. The word 'eustatic' was first used by Eduard Suess to describe changes in sea level "which take place at an approximately equal height, whether in a positive or negative direction, over the whole globe" (Suess, 1906). Today, it is used to describe the relationship between global ice volume change and global mean sea-level change, but the conversion is not straightforward (Milne et al., 2002), and the term 'eustatic' has been used inconsistently in the literature (Lambeck et al., 2001). In light of this, the most recent IPCC report does not use the term 'eustatic', but instead adopts the term 'barystatic' to define global mean sea-level changes resulting from a change in the mass of the ocean (IPCC, 2013). By accounting for the time-dependence of ocean area in Eq. (3), we acknowledge the fact that global mean sea-level change will depend on the rheology of the solid Earth. Inclusion of this dynamic effect, along with consideration of rotational feedback (Sect. 2.2.2),

makes the sea-level equation a non-linear equation. The final two terms of Eq. (3) are the spatial average over the ocean (indicated by the overbar) of the spatially-varying terms in Eq. (2). These final terms must be subtracted because although the mean of the spatially-varying terms will be zero when integrated over the whole of Earth's surface, the mean will not necessarily be zero when integrated over the ocean; hence a uniform shift is applied to conserve mass.

The following two sections describe recent extensions to the sea-level equation, and a description of how it has been used to provide confirmation of several global-scale processes that were hypothesized during the 19[th] and early 20[th] centuries.

### 2.2.2 Extensions to the sea-level equation

The original statement of the sea-level equation by Farrell and Clark (1976) did not account for temporal variations in ocean
area, which can arise via two processes (Figure 3). First, since the ocean is not typically bounded by vertical cliffs, a rise or fall in sea level at a particular location will result in onlap or offlap, and hence an increase or decrease in the area of the ocean, respectively. This issue was first addressed by Johnston (1993). Secondly, during past glacial periods all the major ice sheets grew beyond the confines of the continent on which they were initially situated, expanding into the ocean and forming large areas of marine-grounded ice. Temporal variations in the extent of a marine-grounded ice sheet will alter the ocean area over
which meltwater can be redistributed. The treatment of marine-grounded ice within the sea-level equation was first discussed by Milne (1998), and a detailed description of how to implement shoreline migration due to both processes is given in Mitrovica and Milne (2003).

An additional extension to the sea-level equation involves the treatment of rotational feedback (Figure 4). It is clear that since
GIA alters the distribution of mass throughout the Earth system this will perturb the magnitude and direction of Earth's rotation vector (e.g. Nakiboglu and Lambeck, 1980; Sabadini et al., 1982; Wu and Peltier, 1984). These changes will, in turn, affect a number of processes associated with GIA: changing the Earth's rotation vector will instantaneously alter the shape of the sea surface, i.e. the shape of the geoid (Milne and Mitrovica, 1998), and cause elastic deformation, while over longer timescales it will excite viscous deformation of the solid Earth (Han and Wahr, 1989), thus further altering the shape of the geoid (Figure
4a). Over both timescales these mechanisms result in a long-wavelength change in the distribution of water across the ocean and this will excite additional solid Earth deformation, thus further altering the rotational state of the Earth. These feedbacks were first implemented within the sea-level equation by Milne and Mitrovica (1998), and a number of important updates to the theory have been made in recent years (Mitrovica et al., 2005; Mitrovica and Wahr, 2011; Martinec and Hagedoorn, 2014).

**2.2.3 Confirmation of early theories and implications for the interpretation of sea-level records**

Solutions to the sea-level equation reflect processes that were originally described by Jamieson (1865), Croll (1875), Gilbert (1885b), Woodward (1888), and Nansen (1921). Temporal variations in water depth arise due to changes in the total mass of the ocean (as described by Croll (1875)) and the shape of its two bounding surfaces; the sea surface (as proposed by Woodward

(1888)) and the solid Earth (as proposed by Jamieson (1865) and Gilbert (1885b)). Furthermore, as suggested by Nansen (1921), the shape of these bounding surfaces will continue to evolve even during periods of constant global ice mass due to the time dependent nature of the solid Earth response to surface loading. Observations of relative sea-level change therefore require careful interpretation, particularly if they are to be used to determine changes in global ice volume.

The magnitude of sea-level change in the far-field of the major ice sheets has long been used to constrain changes in global ice volume (e.g. Fairbanks, 1989; Fleming et al., 1998; Milne et al., 2002), but this approach is complicated by the fact that the location at which 'eustatic' or mean sea-level change is recorded will vary over time (Milne and Mitrovica, 2008). It is clear that sea-level change in the *near-field* of an ice sheet will reflect perturbations to the shape of the geoid and the solid Earth due to the presence, and loading effect, of the evolving ice sheet as well as changes in total ocean mass (e.g. Shennan et al., 2002), but *far-field* records of sea-level change will also be biased by long-wavelength, spatially-varying processes associated with GIA. The most important of these processes are outlined below:

- **Meltwater fingerprints (building on theory developed by Woodward, 1888)**: Sea-level change associated with the addition of meltwater to the ocean will be spatially-variable (Milne et al., 2009); a decrease in water depth will be recorded in the near-field of a melting ice sheet due to solid Earth rebound and a fall in the height of the geoid in response to the decrease in ice mass (Figure 2a). Consequently, the increase in water depth far from the melting ice sheet will be greater than the global mean. Predictions of this 'fingerprint' of sea-level change (Plag and Jüttner, 2001) associated with different ice-sheet melt scenarios have been used to distinguish between melt sources during past and present periods of rapid sea-level change (Mitrovica et al., 2001; Clark et al., 2002; Hay et al., 2015).

- **Ocean syphoning (originally hypothesized by Nansen, 1921)**: In the same way that rebound in response to ice mass loss can persist for many thousands of years, subsidence of peripheral bulge regions also continues long after the ice sheets have melted. These peripheral bulge regions surround the former ice sheets and are typically located offshore, and hence their collapse acts to increase the capacity of the ocean basins (Figure 2b). In the absence of significant changes in ocean volume, peripheral bulge collapse will result in a fall in absolute sea level (the height of the sea surface relative to the centre of the Earth) even though global mean water depth will be unchanged. This 'ocean syphoning' effect explains why mid-Holocene sea-level highstands are observed across many equatorial regions (Mitrovica and Peltier, 1991b; Mitrovica and Milne, 2002), and it must also be accounted for when interpreting contemporary measurements of global sea-level change derived from satellite altimetry (Tamisiea, 2011). At sites located on a subsiding peripheral bulge, relative sea-level rise will occur throughout an interglacial period, even if global ice volumes remain roughly constant (Lambeck et al., 2012).

- **Continental levering**: During the LGM lowstand many continental shelves were sub-aerially exposed. Loading by the ocean during the subsequent sea-level rise will have caused the newly-submerged continental shelves to be flexed downwards and the margins of the continents to be flexed upwards (Walcott, 1972). This 'continental levering' effect must be accounted for when interpreting sea-level records recovered from regions adjacent to extensive continental shelves. In particular, it should be noted that coastlines orientated perpendicular to the continental shelf break will experience differential amounts of uplift (e.g. Lambeck and Nakada, 1990; Clement et al., 2016).

## 2.3 Numerical methods used to model GIA

### 2.3.1 Representation of the solid Earth

In order to calculate the solid Earth response to surface load change over glacial timescales the Earth is commonly assumed to be a linear Maxwell viscoelastic body (Peltier, 1974), although a number of studies alternatively adopt a power-law approach (Wu, 1998). The spatially-variable, time-dependent response of a Maxwell body to surface load change can be calculated using viscoelastic Love numbers (building on the work of Love, 1909), which define the response of a spherically-symmetric, self-gravitating, viscoelastic sphere to an impulse point load (Peltier, 1974; Wu, 1978; Han and Wahr, 1995). The Love numbers reflect the assumed viscosity profile of the mantle, which must be defined a priori. Alternatively, if a power-law approach is used, the problem becomes non-linear and the Love number approach cannot be used. Instead, the effective viscosity of the mantle will depend on the stress field throughout the mantle, which depends on surface load change. The non-linear stress-strain relationships that form the basis of the power-law approach are based on the results of laboratory experiments that seek to understand the controls on deformation within the mantle (Hirth and Kohlstedt, 2003). For both approaches the elastic and density structure throughout the Earth must be defined (e.g. Dziewonski and Anderson, 1981), and the deformation of the whole Earth must be considered if the sea-level equation is to be solved (recall that the sea-level equation solves for global meltwater distribution).

In a GIA model the lithosphere is typically represented by an elastic layer or a viscoelastic layer with viscosity high enough to behave elastically on the timescale of a glacial cycle (tens of thousands of years) (e.g. Kuchar and Milne, 2015). The thickness of this layer influences the wavelength of deformation (Nield et al., 2018), while the viscosity of the mantle controls the rate of deformation. It therefore follows that the rheological properties of the Earth may be inferred from observations (Figure 1) that reflect land uplift or subsidence in response to ice and ocean load change (e.g. Lambeck et al., 1998; Paulson et al., 2007a; Lambeck et al., 2014; Peltier et al., 2015; Lau et al., 2016; Nakada et al., 2016). However, in reality, poor data coverage, uncertainties associated with the ice load history, and spatial variations in Earth rheology make it difficult to uniquely determine an optimal solution for Earth properties such as lithosphere thickness or mantle viscosity. To overcome this, some studies consider multiple geodynamic processes when seeking to constrain mantle rheology (e.g. Mitrovica and Forte, 2004), while others use independent data sets to define the rheological properties of the Earth. As an example, seismic wave speeds

can be related to the temperature distribution in the mantle, which in turn may be related to mantle viscosity (Ivins and Sammis, 1995). This approach is discussed in more detail in Sect. 3.2.

### 2.3.2 Modelling approaches

When considering a spherically-symmetric Earth with linear rheology, the sea-level equation is most commonly solved using
a pseudo-spectral approach (e.g. Mitrovica and Peltier, 1991b; Mitrovica and Milne, 2003; Kendall et al., 2005; Spada and Stocchi, 2006; Adhikari et al., 2016). However, finite element (e.g. Wu and van der Wal, 2003; Zhong et al., 2003; Paulson et al., 2005; Dal Forno et al., 2012), spectral-finite element (e.g. Martinec, 2000; Tanaka et al., 2011), and finite volume (e.g. Latychev et al., 2005b) approaches have also been used, while approaches that use the adjoint method are under development (Al-Attar and Tromp, 2014; Martinec et al., 2015). The equations used to represent solid Earth deformation may differ between
these approaches, and in particular the finite element approach was originally developed to permit consideration of power-law rheology (Wu, 1992). A description of the different methods used to determine the response of the solid Earth to surface loading is given in the GIA benchmarking study of Spada et al. (2011). In all cases, an iterative approach is required to determine a gravitationally self-consistent solution to the sea-level equation since the time-dependent change in ocean loading is not known a priori.


A number of studies have sought to model the solid Earth component of GIA without solving the sea-level equation. These are often regional studies, where the focus is on determining the solid Earth response to local ice-load change (e.g. Auriac et al., 2013; Mey et al., 2016), and the effect of global ocean change is less important and has a negligible effect on the results. Focusing on a regional rather than a global domain allows the surface load to be modelled at high resolution (e.g. Nield et al.,
2014), or lateral variations in Earth structure to be incorporated (e.g. Kaufmann et al., 2000; Kaufmann et al., 2005; Nield et al., 2018), while maintaining computational efficiency. A finite element approach is often used, and for domains up to the size of the former Fennoscandian ice sheet the sphericity of the Earth can be neglected (Wu and Johnston, 1998), allowing a 'flat-earth' approximation to be used.

In a few cases GIA models have been extended to explore the potential for GIA-related stress change to trigger earthquakes (e.g. Spada et al., 1991; Wu and Hasegawa, 1996; Steffen et al., 2014c, d; Steffen et al., 2014b; Brandes et al., 2015). The majority of these studies use either a 2D or 3D finite element approach that includes an elastic upper layer and a viscoelastic mantle. Within the model the stress field associated with GIA is combined with the background tectonic stress field, and a Coulomb failure criterion is implemented on pre-existing fault planes to identify faulting events. Models have been used to
calculate the likely magnitude and timing of slip on a range of different orientations of faults in response to different ice sheet sizes (Steffen et al., 2014b) as well as the resulting change in the regional stress field (Brandes et al., 2015).

## 2.4 Data

A fundamental component of GIA modelling is the use of data to constrain unknown factors associated with the ice history
and Earth rheology (Figure 1). Different data have different roles. For example, dated geomorphological evidence for past ice
extent can be used to define the surface load history, while observations relating to solid Earth deformation, such as relative
sea-level indicators or GPS data, can be used to tune the rheological model. There exist strong trade-offs between the timing
and the magnitude of past surface load change (Figure 5a), as well as between the load history and the assumed rheology
(Figure 5b). One way to address this non-uniqueness is to independently constrain ice history and Earth rheology outside the
confines of the GIA model. Alternatively, data sets that are sensitive to both factors, such as observations relating to past sea-
level change, provide very powerful constraints on the coupled problem (e.g. Lambeck et al., 1998). Future work should focus
on collecting new data from locations that are optimally sensitive to the details of ice history or Earth rheology (Wu et al.,
2010; Steffen et al., 2012a; Steffen et al., 2014a). In all cases where data are used to tune a GIA model, it is important to assess
whether there are unmodelled processes reflected in the data that may bias the results, and care must be taken to assign realistic
errors. The key data sets used in studies of GIA are briefly described below.

### 2.4.1 Relative sea-level data

A sea-level indicator is a piece of evidence that provides information on past sea level. In order to be compared with GIA
model output, the age and current elevation of a sea-level indicator must be known (including associated uncertainties), as well
as the relationship between the sea-level indicator and mean sea level. Past relative sea-level change will be preserved in the
geological record as a change in the position of the shoreline or a change in water depth. Past shoreline change can be
reconstructed by identifying the time at which a particular location was inundated by, or isolated from, the ocean (Figure 3).
Such information can be derived from microfossil analysis of the sediment contained within isolation basins (lakes that were
previously connected to the ocean, or former lakes that are now drowned) (e.g. Watcham et al., 2011), or by determining the
age of an abandoned beach ridge, after accounting for the offset between the beach ridge and mean sea level in the modern
setting. In some cases, sea-level indicators may only indicate whether a particular location was previously above or below sea
level. For example, archaeological artefacts typically provide an upper bound on contemporaneous sea level, while the presence
of any type of *in situ* marine material provides a lower bound on past sea level. More specifically, if a fossil shell or coral is
found still in its growth position (either above or below present sea level), and its living-depth range is known, this can be used
to determine past water depths (e.g. Deschamps et al., 2012). Although, note that temporal variations in local conditions, e.g.
changes in water properties or tidal range, can alter the depth at which a particular species can survive (Hibbert et al., 2016).
At higher latitudes, reconstructions of saltmarsh environments have proved very useful for determining not only past changes
in water depth, but also more subtle information relating to whether sea level was rising or falling in the past (Barlow et al.,
2013). Finally, if past shorelines can be continuously reconstructed over length scales of a few kilometres or more, then the
subsequent warping of these contemporaneous surfaces provides a powerful constraint on GIA (McConnell, 1968).

Contemporary sea-level change can be determined by analysing historical tide gauge data and/or the altimetry record (e.g. Church and White, 2011). Much of the observed spatial variation will be due to steric changes, but if this can be accounted for, then the remaining pattern of sea-level change provides information on both past and present ice-sheet change (Hay et al., 2015).


### 2.4.2 Ice extent data

Data relating to past ice extent, thickness, and flow direction all contribute useful information to ice-sheet reconstructions, with the latter providing an indication of past ice-sheet dynamics, and hence the location of former ice domes (e.g. Margold et al., 2015). Terrestrial and marine geomorphological features that must have formed at the margin of a former ice sheet, such

as moraines, grounding zone wedges, or deposits relating to ice-dammed lakes, can be used to build a picture of past ice extent if the age of the features can be precisely dated. Indeed, a series of snapshots of past ice-sheet extent have been constructed from geomorphological data for the Laurentide, British-Irish, and Fennoscandian ice sheets (Dyke et al., 2003; Clark et al., 2012; Hughes et al., 2016). In contrast, determining past ice *thickness* over large spatial scales is more difficult. Field-based reconstructions of past ice thickness typically rely on cosmogenic exposure dating to determine when, and to what depth,

mountain ranges in the interior of a former ice sheet were last covered by ice (Ballantyne, 2010). Care must be taken when interpreting such information because complex topography will perturb the local ice flow, with the result that local ice thickness fluctuations may not represent regional scale ice sheet thickness change. Another issue that must be taken into consideration is the fact that often only evidence relating to the last glacial advance will be preserved, with evidence relating to earlier fluctuations typically having been destroyed due to the erosive nature of ice.


The task of determining the history of an ice sheet that is still present is more difficult, since any evidence relating to a smaller-than-present ice sheet will be obscured. Such a configuration can be inferred if moraines are truncated by the current ice sheet or if contemporary ice-sheet retreat exposes organic material that has been preserved beneath the ice – such material can be dated to determine when it was overrun by ice (Miller et al., 2012). An alternative approach that should be pursued is the

recovery of geological samples from beneath the current ice sheets; a number of techniques (e.g. cosmogenic exposure dating, optically stimulated luminescence dating) can be used to determine how long such samples have been covered by ice. Finally, sampling of ice cores extracted from the ice sheet can provide an indication of past ice thickness, via the analysis of either the gas bubbles preserved in the ice or the isotopic composition of the ice itself (Parrenin et al., 2007).

Due to the sparse nature of ice extent data, numerical ice-sheet models are often used to 'fill the gaps' between field constraints, drawing on the physics of ice flow to determine the likely configuration and thickness of past ice sheets (e.g. Simpson et al., 2009; Whitehouse et al., 2012a; Tarasov et al., 2012; Gomez et al., 2013; Briggs et al., 2014; Lecavalier et al., 2014; Gowan et al., 2016; Patton et al., 2017). See Sect. 3.1 and Sect. 4.2.1 for further discussion of the role of ice-sheet modelling within studies of GIA.

**2.4.3 Surface deformation data**

A number of geodetic data sets are used to quantify surface deformation associated with GIA (King et al., 2010), including Global Positioning System (GPS) data, Interferometric Synthetic Aperture Radar (InSAR) data, and a combination of altimetry and tide gauge data (Nerem and Mitchum, 2002; Kuo et al., 2004). The full potential of InSAR has yet to be realized in the field of GIA – current studies are limited to Iceland (Auriac et al., 2013) – but there is a long tradition of GPS data being used to constrain GIA models. These data must be corrected for signals associated with the global water cycle, atmospheric effects, and local processes associated with tectonics or sediment compaction (King et al., 2010). In areas where non-GIA signals are well constrained and there is a dense network of measurements, such as across North America (Sella et al., 2007) or Fennoscandia (Lidberg et al., 2007), GPS data have successfully been used to calibrate GIA models (e.g. Milne et al., 2001; Milne et al., 2004; Lidberg et al., 2010; Kierulf et al., 2014; Peltier et al., 2015). However, in regions where contemporary ice mass change also contributes to present-day solid Earth deformation, it becomes difficult to disentangle contributions from past and present ice-sheet change (Thomas et al., 2011; Nield et al., 2014). Horizontal GPS rates are often more precise than vertical rates by an order of magnitude (King et al., 2010), but before they can be compared with GIA model output the velocity field due to plate motion must be removed. This is non-trivial, since neither plate motion nor GIA are perfectly known (King et al., 2016). It has long been known that horizontal deformation in response to surface loading can be strongly perturbed by the presence of lateral variations in Earth rheology (Kaufmann et al., 2005), and future work should make use of this opportunity to better understand the Earth structure in regions affected by GIA (e.g. Steffen and Wu, 2014).

Geodetic information is typically provided on a reference frame whose origin is located at the centre of mass of the entire Earth system (e.g. ITRF2008; Altamimi et al., 2012) while GIA model predictions are typically provided on a reference frame whose origin lies at the centre of mass of the solid Earth. Reference frame differences must therefore be accounted for when comparing model output with GPS data, along with uncertainties associated with realization of the origin of the reference frame (King et al., 2010).

**2.4.4 Gravity data**

Between 2002 and 2017, repeat measurements of the Earth's gravity field by the Gravity Recovery and Climate Experiment (GRACE) satellites allowed temporal variations in the distribution of mass throughout the cryosphere, the atmosphere, the oceans, and the solid Earth to be quantified (e.g. Wouters et al., 2014). One of the principle drivers of solid Earth deformation is GIA, and across previously-glaciated regions that are now ice-free, GRACE data (and measurements of the static gravity field by the 'Gravity field and steady-state Ocean Circulation Explorer', GOCE) have been used to quantify the magnitude and spatial pattern of the local GIA signal (e.g. Tamisiea et al., 2007; Hill et al., 2010; Metivier et al., 2016), past ice thickness (e.g. Root et al., 2015), and local viscosity structure (e.g. Paulson et al., 2007a). However, in areas where an ice sheet is still present variations in the local gravity field will reflect the solid Earth response to both past and present ice mass change, as

well as contemporary changes to the mass of the ice sheet itself (Wahr et al., 2000) and non-GIA-related mass redistribution.
In this situation, a joint approach to solving for GIA and contemporary ice mass change is necessary, often via the combination of GRACE data with other data sets (see Sect. 3.4) (e.g. Sasgen et al., 2007; Riva et al., 2009; Ivins et al., 2011; Groh et al., 2012; Gunter et al., 2014; Martin-Espanol et al., 2016b).

On a more local scale, absolute gravity measurements have been used to study GIA (e.g. Peltier, 2004; Steffen et al., 2009;
Mazzotti et al., 2011; Memin et al., 2011; Sato et al., 2012), while the relationship between surface gravity change and uplift rates can be employed to constrain GIA in regions where ice history and Earth structure are poorly known (e.g. Wahr et al., 1995; Purcell et al., 2011; van Dam et al., 2017; Olsson et al., 2015).

### 2.4.5 Independent constraints on solid Earth properties

The rheology of the mantle and the thickness of the lithosphere are often inferred by comparing GIA model output with observations that reflect past and present rates of solid Earth deformation, such as GPS time series or records of past relative sea-level change (e.g. Lambeck et al., 1998; Whitehouse et al., 2012b; Argus et al., 2014). However, GIA model predictions will be sensitive to the assumed ice history, and therefore it can be useful to draw on independent information to constrain properties of the solid Earth.


For the purposes of GIA, the elastic and density structure of the Earth is assumed to follow that of the PREM (Dziewonski and Anderson, 1981), which is derived from seismic data. Lithosphere thicknesses can be inferred from inversions of gravity or seismic data, or via thermal modelling, although it should be noted that the apparent thickness of the lithosphere will depend on the timescale of the loading (Watts et al., 2013), and hence values derived by considering e.g. the seismic thickness of the
lithosphere, or its elastic thickness over geological timescales, will not be relevant for GIA. Finally, mantle viscosities can be independently estimated via a number of approaches, including consideration of processes associated with mantle convection (e.g. Mitrovica and Forte, 2004), or via the conversion of seismic velocity perturbations into mantle viscosity variations (e.g. Ivins and Sammis, 1995; Wu and van der Wal, 2003; Latychev et al., 2005b; Paulson et al., 2005).

The long wavelength response of the solid Earth to surface mass redistribution since the LGM principally depends on the viscosity of the lower mantle, and it results in changes to the oblateness of the solid Earth ($\dot{J}_2$), the position of the geocenter, and the orientation of the rotation pole (Figure 4). If these processes can be quantified (e.g. Gross and Vondrak, 1999; Cheng and Tapley, 2004) they can be used to place constraints on lower mantle viscosity (e.g. Paulson et al., 2007b; Mitrovica and Wahr, 2011; Argus et al., 2014; Mitrovica et al., 2015). However, it should be noted that these large-scale processes will also
be affected by contemporary surface mass redistribution, for example, associated with melting of the polar ice sheets (Adhikari and Ivins, 2016).

**2.4.6 Stress field**

Unloading of the solid Earth during deglaciation alters the regional stress field, and can trigger glacially-induced faulting
(Arvidsson, 1996; Lund, 2015). However, it is not straightforward to infer past changes in surface loading from the regional
faulting history because although deglaciation can trigger faulting, GIA-induced stress changes are probably only capable of
triggering slip on pre-existing faults (Steffen et al., 2014b), with the fault expression reflecting the underlying tectonic stress
field as well as the GIA-related stress field (Steffen et al., 2012b; Craig et al., 2016). Glacial loading is thought to promote
fault stability (Arvidsson, 1996; Steffen et al., 2014c), with the main period of fault activation taking place soon after the end
of glaciation, during the period of maximum rebound (Wu and Hasegawa, 1996; Steffen et al., 2014b). The timing of faulting
can therefore provide some insight into the timing of ice unloading, and potentially also the rheological properties of the mantle
(Brandes et al., 2015). It is more difficult to draw conclusions about the spatial history of the ice sheet from the distribution of
faulting because modelling suggests that only small stress changes are required to trigger seismicity (Steffen et al., 2014b),
and so glacially-induced earthquakes may be distributed over a large area that does not necessarily reflect the spatial extent of
the former ice sheet (Brandes et al., 2015). Finite-element modelling of glacially-induced faulting indicates that the magnitude
of fault slip is primarily governed by shallow Earth properties and fault geometry (Steffen et al., 2014d; Steffen et al., 2014b).

**2.5 Significant results**

Over the past 40 years, GIA modelling has played a central role in advancing our understanding of the rheology of the Earth,
the history of the global ice sheets, and the factors controlling spatially-variable sea-level change. Key results are briefly
outlined below.

**2.5.1 Mantle viscosity**

GIA modelling is one of the principle approaches used to determine mantle viscosity. A range of global and regional studies
indicate that the mean viscosity of the upper mantle lies in the range $10^{20}$ - $10^{21}$ Pa s, while the viscosity of the lower mantle
is less tightly constrained to lie in the range $10^{21}$ - $10^{23}$ Pa s (e.g. Mitrovica, 1996; Lambeck et al., 1998; Milne et al., 2001;
Peltier, 2004; Bradley et al., 2011; Steffen and Wu, 2011; Whitehouse et al., 2012b; Lambeck et al., 2014; Lecavalier et al.,
2014; Peltier et al., 2015; Nakada et al., 2016). It is generally agreed that the viscosity of the lower mantle is greater than that
of the upper mantle, but the magnitude of the increase across this boundary continues to be the subject of significant discussion
(e.g. Mitrovica and Forte, 2004; Lau et al., 2016; Caron et al., 2017). GIA modelling can be used to solve for the depth-
dependent viscosity profile of the mantle, but it is important to assess the resolving power of the constraining data sets when
considering the accuracy and uniqueness of the results (Mitrovica and Peltier, 1991a; Milne et al., 2004; Paulson et al., 2007b).
Finally, GIA modelling of the response to recent (centennial-scale) ice mass change has been used to identify a number of
localized low viscosity regions (<$10^{20}$ Pa s) where deformation occurs over a much shorter timescale, e.g. in Iceland (Pagli et

al., 2007; Auriac et al., 2013), Alaska (Larsen et al., 2005; Sato et al., 2011), Patagonia (Ivins and James, 2004; Lange et al., 2014), and the Antarctic Peninsula (Simms et al., 2012; Nield et al., 2014).

### 2.5.2 Ice-sheet change

GIA modelling has been used to infer past global ice volumes, primarily via the comparison of low latitude relative sea-level
records with GIA model output. Estimates of global ice volume during three key periods are summarized below:

i) Global ice volume change since the LGM is thought to be equivalent to ~130 m sea-level rise (e.g. Peltier, 2004; Lambeck et al., 2014). The LGM lowstand occurred ~21 ka, and melting of the Laurentide and Fennoscandian ice sheets was largely complete by 7 ka. Small magnitude ice volume changes subsequent to this time are less well constrained (Lambeck et al., 2014; Bradley et al., 2016).

ii) Combining GIA modelling with a probabilistic approach, Kopp et al. (2009) find that global ice volumes during the Last Interglacial (~125 ka) were at least 6.6 m smaller than present (95% probably; magnitude expressed as sea-level equivalent). Uncertainty associated with the interpretation and dating of sea-level indicators (Rovere et al., 2016; Düsterhus et al., 2016b) and neglect of non-GIA processes (Austermann et al., 2017) hampers our ability to more precisely reconstruct changes in global ice volume during this period.

iii) Considering even earlier warm periods, Raymo et al. (2011) demonstrate that the scatter in Pliocene (~3 Ma) shoreline elevations (typically found between 10 and 40 m above present sea level) can partly be explained by GIA. However, in order to reconstruct global ice volumes at this time, the complicating effects of tectonics, dynamic topography, and sediment compaction must be accounted for (Rovere et al., 2014).

In addition to constraining global ice volumes, comparison of GIA model output with a range of data sets has been used to reconstruct the past configuration of individual ice sheets, including the Fennoscandian (Lambeck et al., 1998; Lambeck et al., 2010), British-Irish (Lambeck, 1995; Peltier et al., 2002; Bradley et al., 2011), Laurentide (Tarasov et al., 2012; Simon et al., 2016; Lambeck et al., 2017), Greenland (Tarasov and Peltier, 2002; Simpson et al., 2009; Lecavalier et al., 2014), and Antarctic (Whitehouse et al., 2012a; Whitehouse et al., 2012b; Ivins et al., 2013; Gomez et al., 2013; Argus et al., 2014; Briggs et al., 2014) ice sheets. Due to a lack of constraining data, there are often large discrepancies between different ice-sheet
reconstructions. Global ice-sheet reconstructions also exist (Peltier, 2004; Peltier et al., 2015; Lambeck et al., 2014), but the important question of whether the total volume of the individual ice sheets is sufficient to account for the magnitude of the LGM lowstand remains unresolved (Clark and Tarasov, 2014).

Finally, improved quantification of the geodetic signal associated with past ice-sheet change has led to recent improvements in the accuracy of contemporary estimates of ice mass balance, as derived from GRACE or altimetry data (King et al., 2012;

Ivins et al., 2013). However, uncertainty associated with the 'GIA correction' that must be applied to such data sets still poses a significant challenge to studies that seek to reach a fully-reconciled estimate of contemporary ice mass balance, particularly for Antarctica (Shepherd et al., 2012).


### 2.5.3 Sea-level change

Understanding the rate, magnitude, and spatial pattern of past, present and future sea-level change, and linking these changes to climate forcing, is one of the most important questions facing modern society (IPCC, 2013). Important results that have been derived using GIA modelling include:

i)   quantification of the maximum rate of global mean sea-level rise during the last deglaciation (e.g. Lambeck et al., 2014)

    ii)  identification of the potential meltwater source(s) that contributed to rapid sea-level rise during the last deglaciation (e.g. Clark et al., 2002; Gomez et al., 2015a; Liu et al., 2016)

    iii) quantification of the maximum sea-level attained during past warm periods (e.g. Kopp et al., 2009; Dutton et al., 2015)

    iv)  identification of the rate, pattern, and source of historical and contemporary sea-level change (e.g. Riva et al., 2010; Hay
et al., 2015; Rietbroek et al., 2016)

    v)   quantification of the likely pattern of future sea-level change due to ice-sheet change (e.g. Mitrovica et al., 2009; Slangen et al., 2014; Hay et al., 2017)

All of these results draw on the complex relationship between ice sheet-change and spatially-variable sea-level change, as described by the sea-level equation. Significant advances in our understanding of sea-level and ice-sheet change have come
about due to improvements in data availability and GIA modelling capability during the last decade, but persistent uncertainties associated with the GIA correction that must be applied when interpreting gravity, altimetry, tide gauge, or GPS data (Tamisiea, 2011), and ongoing ambiguity associated with the interpretation of palaeo-data, mean than future progress will require input from a diverse range of disciplines.

### 3 Recent Developments

Over the past decade there have been rapid advances in our understanding of how GIA processes can influence other dynamic systems, and an increased awareness of additional factors that must be considered when seeking to constrain or tune a GIA model using independent data sets. New approaches of isolating the GIA signal have also been devised. Four of the most important recent advances are briefly described in this section.

### 3.1 Ice dynamic feedbacks

Inferring the past evolution of the major ice sheets has been a central goal of GIA modelling since Jamieson (1865) first observed that the growth of an ice sheet will depress the land and affect the position of the ocean shoreline. However, it is only recently that glaciologically-consistent ice-sheet reconstructions, i.e. those developed using a numerical ice-sheet model, have begun to be produced for the purposes of GIA modelling (e.g. Tarasov and Peltier, 2002; Tarasov et al., 2012; Whitehouse et al., 2012a). A crucial boundary condition that must be defined when modelling the evolution of a marine-grounded ice sheet

is the water depth of the surrounding ocean. This water depth determines where the ice sheet begins to float, a point known as the grounding line. More importantly, it determines the rate at which ice flows across the grounding line and into the ocean (because ice flux depends on ice thickness (Schoof, 2007)).

       Numerical ice-sheet models are typically run assuming that sea-level change adjacent to an ice sheet will track global mean

sea-level change. Far-field ice melt will indeed cause near-field sea-level rise (Figure 6a), but, due to the effects of GIA, water depth changes will not follow the global mean during near-field ice-sheet change (Figure 2). Nearly 40 years ago, Greischar and Bentley (1980) noted that solid Earth rebound triggered by ice loss from a marine-grounded ice sheet would reduce local water depths, and could promote grounding line advance (Figure 6b). The decreased gravitational attraction of the melting ice sheet also acts to reduce local water depths. Modelling both effects, Gomez et al. (2010) demonstrated that GIA has a stabilizing

effect on the dynamics of a marine-grounded ice sheet, and can prevent or delay unstable grounding line retreat and ice loss.

       Spatially-variable water depth boundary conditions were first used in conjunction with a numerical ice-sheet model, for the purposes of reconstructing past ice-sheet change, by Whitehouse et al. (2012a), who used a priori GIA model output to determine water depths around Antarctica, and hence ice flux across prescribed grounding line positions. Subsequently, Gomez

et al. (2013) and de Boer et al. (2014) and have used fully-coupled ice sheet-GIA models to produce ice-sheet reconstructions that are consistent with spatially-variable sea-level change over time. It is interesting to note that LGM reconstructions for Antarctica generated using coupled models tend to contain 1-2 m less ice (expressed as sea-level equivalent) than reconstructions generated using uncoupled models (de Boer et al., 2017). If the coupled model results are robust this makes it difficult to account for the global mean sea-level lowstand during the LGM (Clark and Tarasov, 2014).


       Considering future ice-sheet change, Adhikari et al. (2014) have used one-way coupling to quantify the impact of ongoing GIA on Antarctic ice dynamics up to 2500 AD, while Gomez et al. (2015b) and Konrad et al. (2015) have used coupled models to investigate the long-term evolution of the ice sheet, finding that GIA-related feedbacks have the potential to significantly limit, or even halt, future ice loss if the upper mantle viscosity beneath West Antarctica is low enough for rapid rebound to be

triggered. A crucial factor in determining the stability of an ice sheet is the resistance provided by the surrounding floating ice shelves. If rebound is fast enough for the ice shelves to re-ground on submerged topographic highs, forming ice rises (Matsuoka

et al., 2015), this significantly increases the chance of ice-sheet stabilization or even re-growth (Kingslake et al., in press). Uncertainties associated with the bathymetry and the upper mantle viscosity beneath the Antarctic and Greenland ice sheets currently present the greatest barriers to accurately quantifying the degree to which GIA-related feedbacks have the potential
to limit future ice loss from these regions.

## 3.2 Lateral variations in Earth rheology, and non-linear rheology

GIA models traditionally assume the Earth behaves as a linear Maxwell viscoelastic body with a viscosity profile that varies in the radial direction only and stays constant with time (e.g. Peltier et al., 2015). A number of studies have made use of a bi-viscous Burgers rheology within a radially-varying framework, in which mantle deformation is dominated by the behaviour
of two viscosities linearly relaxing over different timescales (Yuen et al., 1986; Caron et al., 2017). However, an increasing number of studies are making use of a framework that can accommodate three-dimensional variations in mantle viscosity (e.g. Ivins and Sammis, 1995; Martinec, 2000; Latychev et al., 2005b; Kaufmann et al., 2005; Paulson et al., 2005; Steffen et al., 2006; A. et al., 2013), possibly defined via use of a non-linear creep law – where the viscosity depends on the time-varying stress field (Wu et al., 2005; van der Wal et al., 2013). The effect of including plate boundaries (Klemann et al., 2008) – which
affect the horizontal transmission of stress – and variations in the thickness and rheology of the lithosphere have also been explored (Latychev et al., 2005a; Wang and Wu, 2006a; Wang and Wu, 2006b; Kuchar and Milne, 2015). The development of these '3D GIA models' are motivated by (i) convincing evidence for strong lateral variations in rheological properties beneath some regions, including Antarctica (Heeszel et al., 2016); and (ii) the demonstration that consideration of lateral variations in rheology is required to correctly model horizontal deformation (Kaufmann et al., 2005).


It is important to question whether such increased model complexity is necessary. Whitehouse et al. (2006) showed that inclusion of 3D Earth structure perturbs uplift rate predictions across Fennoscandia by an amount greater than current GPS accuracy, with significant implications for inferences of past ice-sheet history, while Kendall et al. (2006) demonstrated that relative sea-level change predictions will be biased by >0.2 mm/yr at ~150 global tide gauge sites if 3D Earth structure is
neglected, with maximum differences exceeding several mm/yr. Since solid Earth deformation depends on both the surface load history and Earth rheology, non-uniqueness is a problem when solving for these two unknowns. If a GIA model is tuned to fit GIA-related observations (e.g. uplift rates, sea-level records) without accounting for lateral variations in rheology, the resulting ice-sheet reconstruction is likely to be biased. For example, past ice thickness change is likely to be overestimated in regions where the local mantle viscosity is weaker than assumed by the model (Figure 5b). Similarly, global ice volumes may
be incorrectly inferred if viscosity variations are ignored at far-field sea-level sites (Austermann et al., 2013). If the past ice history of a region has been independently determined then neglect of lateral variations in Earth structure will lead to bias in predictions of the GIA signal, and hence bias in estimates of contemporary ice sheet mass balance, potentially on the order of 10's Gt/yr (van der Wal et al., 2015). Furthermore, models that consider the coupled evolution of the ice sheet-solid Earth

system (see Sect. 3.1) will be highly sensitive to the underlying viscosity field (Gomez et al., 2015b; Konrad et al., 2015; Pollard et al., 2017; Gomez et al., 2018).

A range of approaches are used to define spatial variations in Earth rheology, but most rely on deriving a temperature field from a seismic velocity model (e.g. Ritsema et al., 2011), from which a viscosity field is derived (e.g. Ivins and Sammis, 1995; Latychev et al., 2005b). This derivation is not straightforward, and in particular, compositional effects must be accounted for when interpreting seismic velocity perturbations in terms of temperature perturbations (Wu et al., 2013). If a power-law approach is used, grain-scale deformation of mantle material is described by diffusion and dislocation creep through the use of a non-linear relationship where the strain rate depends on stress to some power. This power is thought to be 1 for diffusion creep – i.e. a linear response to forcing – but ~3.5 for dislocation creep (Hirth and Kohlstedt, 2003). This has implications for the inferred viscosity of the mantle: if dislocation creep is important, i.e. there is a non-linear relationship between stress and strain rate, then the effective viscosity of the mantle will depend on the von Mises stress. Since the von Mises stress depends on the evolution of the ice/ocean surface load it follows that the effective viscosity will be time dependent. Inputs to the power-law relationship include grain size, water content, and melt content, as well as temperature, although a lack of direct observational data for most of these parameters mean that values derived in laboratory experiments are often adopted (Hirth and Kohlstedt, 2003; Burgmann and Dresen, 2008; King, 2016). Significant further work is needed to better quantify the viscosity distribution throughout the mantle, and to determine the spatial resolution at which viscosity variations must be resolved to accurately reflect the global GIA process (Steffen and Wu, 2014).

### 3.3 Sedimentary isostasy

The isostatic response to sediment erosion and deposition, on glacial and longer timescales, has long been considered in studies of onshore and offshore crustal deformation, associated with both fluvial and glacial systems. However, the impact of sediment redistribution on Earth's gravitational and rotational fields, and the consequent effect on sea-level, has only recently been considered. Dalca et al. (2013) were the first to incorporate the gravitational, deformational, and rotational effects of sediment redistribution into a traditional GIA model (Figure 7a). The resulting theory has been used to demonstrate that the impact of sediment erosion and deposition, associated with both fluvial and glacial systems, can alter relative sea-level by several metres over the course of a glacial cycle and rates of present-day deformation by a few tenths of a mm/yr (Wolstencroft et al., 2014; Ferrier et al., 2015; van der Wal and IJpelaar, 2017; Kuchar et al., 2017). Although the magnitude of the perturbation due to sediment loading is small, it is greater than the precision of modern geodetic methods, and hence has the potential to bias contemporary estimates of sea-level change (Ferrier et al., 2015; van der Wal and IJpelaar, 2017). Perhaps the most important finding of these preliminary studies is the observation that in order for relative sea-level indicators to be used to constrain past global ice volumes, they must first be corrected for the effects of both glacial and sedimentary isostasy (Ferrier et al., 2015). As an example, if sediment loading has caused a sea-level indicator to subside subsequent to its formation, this will lead to an overestimation of the magnitude of sea-level rise since the formation of the sea-level indicator.

Sedimentary isostasy is not the only sediment-related process that affects sea level. Wolstencroft et al. (2014) and Kuchar et al. (2017) both found that including the effects of sedimentary isostasy did not bring agreement between model predictions and GPS-derived observations of contemporary land motion around the Mississippi Delta, and they concluded that sediment compaction must play a significant role (Figure 7b). To address this Ferrier et al. (2017) have updated the theory developed by Dalca et al. (2013) so that it accounts for all the competing processes associated with sediment redistribution, including the decrease in water depth due to offshore deposition and the increase in water depth due to subsidence and compaction (Figure 7). This state-of-the-art approach, which uses estimates of sediment porosity and saturation to determine the time-evolving effects of compaction, has recently been used to provide a robust interpretation of sea-level indicators formed during Marine Isotope Stage 3 (~50-37 ka) in the region of China's Yellow River Delta, and hence tighten constrains on global ice volumes at this time (Pico et al., 2016). Water depth changes associated with sediment redistribution and compaction can vary over short spatial scales, and therefore care is needed to interpret individual sea-level indicators, but the modelling approaches described above are well-suited to studying the large-scale impact of sediment redistribution.

In conclusion, both sediment loading and sediment compaction have been demonstrated to have a non-negligible effect on relative sea-level change, and this has far-reaching implications for the interpretation of relative sea-level indicators within the framework of a GIA model (see Sect. 2.4.1). Both effects should be accounted for in future GIA models, although better constraints on the timing and distribution of erosion and deposition, over the last glacial cycle and beyond, are needed before past global ice volumes can be robustly inferred from global sea-level records.

### 3.4 Inverse solutions

The primary motivation for the recent focus on developing GIA models for Greenland and Antarctica has been to permit accurate interpretation of GRACE data, partitioning the mass change time series into contributions from GIA and contemporary ice mass change (King et al., 2012; Ivins et al., 2013). However, turning the problem around, GRACE data can also be combined with altimetry data to determine an estimate of the GIA signal. This is possible because the two data sets have a different sensitivity to GIA. The basic premise of the method is that the GRACE satellites detect the spatial pattern of mass change, which is attributed to the redistribution of ice and solid Earth mass (after accounting for atmospheric and oceanic mass change), whereas altimetry satellites such as ICESat measure the spatial pattern of surface elevation change, which is attributed to a combination of solid Earth deformation and ice volume change. A density model is used to convert the ICESat height measurements to mass, and the two data sets can be used to solve for GIA-related deformation, as shown in the following equation, where $\dot{h}_{GIA}$ is the rate of land elevation change due to GIA, $\dot{h}_{ICESat}$ is the rate of surface elevation change measured by ICESat, $\dot{m}_{GRACE}$ is the rate of total mass change measured by GRACE, $\rho_{rock}$ is the mean density of the upper ~100km of the solid Earth, and $\rho_{surf}$ is the mean density of the ice volume change:

$$\dot{h}_{GIA} = \frac{\dot{m}_{GRACE} - \rho_{surf} \cdot \dot{h}_{ICESat}}{\rho_{rock} - \rho_{surf}}$$  (4)

This method of simultaneously solving for ice mass change and GIA was originally suggested by Wahr et al. (2000). Riva et al. (2009) first implemented it for Antarctica using 5 years of data, and Gunter et al. (2014) improved the method by using a firn densification model instead of assuming a constant density of ice. Further advances were made by Martin-Espanol et al. 725 (2016b), who estimated the GIA signal across Antarctica using a statistical inversion of a suite of geodetic observations, including GRACE, altimetry, InSAR, and GPS data, and a priori information about the wavelength of the GIA signal, while Sasgen et al. (2017) extended the method to account for lateral variations in Earth structure. The advantage of using data inversion to isolate the contemporary GIA signal (Eq. 4) is that the solution is not dependent on the ice loading history or the Earth structure (other than the assumed density of the Earth). Using the GRACE data in a slightly different way, Sasgen et al. 730 (2013) developed a method that systematically searches the forward model parameter space to determine an optimal fit to the GRACE data, using GPS data to provide a further constraint on deformation rates. These data-driven GIA solutions currently exhibit significant variability (Martin-Espanol et al., 2016a), partly due to the large number of assumptions that must be made during processing of the raw data, but they have formal uncertainties attached to them, and hence provide a useful comparison for output derived from process-based forward models of GIA.

**4 Future directions and challenges**

In this final section a number of emerging areas of research are outlined. It is now clear that GIA should be considered within future Earth System modelling efforts, and some of the most exciting developments will come as GIA modelling becomes even better integrated with a range of disciplines.

**4.1 Data interpretation and assimilation**

A central component of GIA modelling has always been the use of independent data sets to either prescribe model inputs, tune model parameters, or test model outputs. As the accuracy and coverage of these data sets improves, along with our understanding of the processes they record, it is important to be aware that it will not be possible to honour all of the constraints provided by the data. A strategy is therefore needed in which the available data are used to develop the most accurate GIA model possible and assign a level of uncertainty to the resulting model output.


Many of the data sets that record GIA processes also record competing processes. Around 50% of contemporary global mean sea-level rise is due to thermosteric effects, reflecting a change in the density of the ocean rather than its mass (IPCC, 2013); the thermosteric contribution to post-LGM global mean sea-level rise has not been quantified. Changes in oceanic or atmospheric circulation will alter the dynamic topography of the ocean surface, while changes in the tidal regime or ocean

conditions (acidity, temperature, opacity) can alter the habitat distribution of species that are used to infer the position of the shoreline. When seeking to reconstruct past ice extent it is important to acknowledge that a change in local ice flow or wind direction can alter the height of the ice-rock contact, while changes in ice surface elevation, as recorded by ice cores, will reflect changes in both ice thickness and the height of the underlying bed (Bradley et al., 2012; Lecavalier et al., 2013). Finally, land motion, as recorded by GPS, will reflect the solid Earth response to contemporary surface mass redistribution (ice, water,

sediment) as well as tectonics and sediment compaction. Over longer timescales (≥100 kyr) land motion also reflects the dynamic response of the lithosphere to mantle convection (Rowley et al., 2013; Austermann et al., 2017). As far as is possible, these competing factors must be accounted for, or appropriate error bars should be attached to the data in a format that can be easily incorporated into a GIA modelling framework (Düsterhus et al., 2016a).

Once a suitable data set has been identified, it can be used to determine an optimal GIA model. Due to data gaps and uncertainties there are often trade-offs between the magnitude and timing of the surface loading history and Earth rheology that can explain the observations (Figure 5). Traditionally, an iterative approach has been used to determine a single, optimal ice history/Earth model combination that provides the best fit to a range of data types (Lambeck et al., 1998; Peltier et al., 2015). Using data assimilation or statistical emulation, in combination with large ensemble modelling, it should be possible to

determine a suite of GIA solutions that provide a reasonable fit to all available constraints. Such an approach has already been pioneered for ice-sheet modelling (Briggs et al., 2013; Pollard et al., 2016), but it requires careful consideration of the probability distribution associated with each piece of constraining data and the method used to score each model run (Briggs and Tarasov, 2013). When applied to GIA modelling, decisions will have to be made on: how to weight different data types, the length scale over which each is relevant, how to treat uncertainties in age and elevation, whether to use raw data or statistical

reconstructions (e.g. Khan et al., 2015), and how to account for additional metadata, such as information on whether sea-level was rising or falling at a particular location. The uncertainty on the resulting GIA estimate should directly reflect uncertainties in the constraining data.

In order to optimize the search through the parameter space associated with ice history and Earth rheology a number of

approaches could be taken. When determining the past evolution of the global ice sheets, a previously-used method involves taking an existing global reconstruction and scaling the thickness of each ice sheet in turn (e.g. Caron et al., 2017). However, building on new understanding of the feedbacks between ice dynamics and GIA (Sect. 3.1), completely new glacial scenarios should also be explored, guided by the output from coupled models. When determining the optimal Earth model, it will be important to allow for the possibility that different regions are characterized by different viscosity profiles, guided by

independent constraints on mantle viscosity variations (Sect. 2.4.5).

There will always be some parts of the parameter space that cannot be constrained. A useful exercise is to determine the locations in which it would be most beneficial to have new data constraints (e.g. Wu et al., 2010), but there are often logistical

barriers to collecting the most useful data. For example, it would be very useful to know the rate of solid Earth rebound beneath the present-day ice sheets and across the ocean floor (although research is underway in this area; see Honsho and Kido, 2017); it would be useful to know the past thickening history of the ice sheets as well as their thinning history; and it would be useful to know how sea level has changed at locations distal from current and past shorelines. In the near future, advances will come through the application of novel analytical techniques in regions where sea-level reconstructions have so far proved challenging, e.g. along mangrove coasts, and the more widespread use of satellite gravity data to constrain the GIA signal where sea-level and GPS data are lacking (Root et al., 2015).

## 4.2 Coupled modelling

Significant advances in understanding GIA have often stemmed from a cross-disciplinary approach. The gravitational theory developed by the mathematician Woodward in the late 19[th] century came about as the result of a question posed by geologists Gilbert and Chamberlin (Woodward, 1888). More recently, observations of a mid-Holocene highstand throughout the low latitudes led to recognition of the ocean syphoning process (Mitrovica and Milne, 2002), while disparate observations relating to the magnitude of the Pliocene highstand led to advances in the modelling of GIA over multiple glacial cycles (Raymo et al., 2011). Sea-level observations, and more recently GPS observations, have often been the motivation for developing new hypotheses relating to the history of the major ice sheets (e.g. Bradley et al., 2015) and the processes governing ice dynamics (e.g. Pollard et al., 2015). Future progress is likely to be made by fully integrating a number of different disciplines, i.e. via coupled modelling.

### 4.2.1 Coupled ice sheet-3D GIA modelling

Coupled ice sheet-GIA modelling has already been discussed in Sect. 3.1, as has 3D GIA modelling (Sect. 3.2), but since the impact of GIA on ice dynamics has been shown to be strongest in locations where mantle viscosity is unusually low (Pollard et al., 2017) it will be crucial to incorporate lateral variations in Earth structure into coupled ice sheet-GIA models (e.g. Gomez et al., 2018) in order to accurately predict the evolution of the many ice sheets that are located above low viscosity regions (e.g. West Antarctica, Patagonia, Alaska, Iceland; see Sect. 4.3). This will be computationally challenging and careful experiment design will be needed to ensure an efficient, yet sufficient, search of the parameter space.

### 4.2.2 Coupled GIA-ocean/atmosphere modelling

The impacts of GIA, i.e. spatially variable perturbations to the sea surface and solid Earth, are not currently considered within atmosphere or ocean circulation models, but are likely to have a non-negligible effect on these systems. Perturbations to the seafloor due to the growth and decay of submerged peripheral bulges (Figure 2b) have been shown to be sufficient to perturb ocean circulation (Rugenstein et al., 2014), while the spatially-variable sea-level change that is predicted to accompany ice melt will impact the global tidal regime (Wilmes et al., 2017). Similarly, changes in surface topography due to solid Earth deformation will affect atmospheric circulation patterns and consequently precipitation patterns. This latter factor has clear

implications for the mass balance of an ice sheet, and has been proposed as an explanation for hysteresis within glacial cycles (Abe-Ouchi et al., 2013). During the LGM the areal extent of the continents will have been greater as a result of GIA. Additional land bridges will have existed, with direct implications for migration pathways and ocean circulation, and in the northern hemisphere the additional land mass may have facilitated ice-sheet expansion or inception. GIA is also able to provide insight into the timing, magnitude and source of freshwater inputs to the ocean during deglaciation. In order to accurately model the palaeo-circulation of the ocean and the atmosphere, some factors associated with GIA should be incorporated into future modelling efforts.

### 4.2.3 Coupled GIA-surface process modelling

GIA-related solid Earth deformation during the last glacial cycle will have affected the routing of palaeo-drainage systems (Wickert et al., 2013), it will have governed the location and extent of ice-dammed lakes (Lambeck et al., 2010; Patton et al., 2017; Lambeck et al., 2017), and it continues to influence coastal evolution (Whitehouse et al., 2007). Accounting for GIA-related factors such as changes to the shape of the land surface and changes to the position of the base level within landscape evolution models will lead to a more complete understanding of Earth surface processes. The full impacts of GIA, including spatially-variable sea-level change and changes to the shape of Earth's gravitational field, should be accounted for in future studies that seek to understand the isostatic response to glaciation, erosion, and sedimentation (e.g. Mey et al., 2016; Moucha and Ruetenik, 2017).

### 4.2.4 GIA-Climate feedbacks

GIA potentially plays a role in modulating climate cycles. Postglacial rebound acts to reduce the pressure in the mantle, and this has been implicated in promoting terrestrial volcanism (Sigmundsson et al., 2010; Schmidt et al., 2013; Praetorius et al., 2016). Huybers and Langmuir (2009) argue that the $CO_2$ release associated with increased volcanism during the last deglaciation may have been sufficient to promote further ice melt, raising the possibility that glacial rebound, $CO_2$ release, and ice dynamics are part of a positive feedback loop. But postglacial rebound is not the only GIA-related process that affects the rate of $CO_2$ release: as the ice sheets wax and wane this alters global sea level, and the resulting stress changes are thought to be sufficient to perturb rates of volcanism (Kutterolf et al., 2013). It was originally assumed that mid-ocean ridge volcanism would be supressed during periods of sea-level rise (i.e. during deglaciation) and this would act to counter the contemporaneous climatic effects of increased terrestrial volcanism (Huybers and Langmuir, 2009). However, the evidence suggests that there is actually a significant delay in the response of the mid-ocean ridge system to a change in seafloor pressure. This can be deduced from records of hydrothermal and magmatic activity along mid-ocean ridges (Crowley et al., 2015; Lund et al., 2016), while models of melt transport through the mantle predict a lag of at least 60 kyr between a fall in sea-level and an increase in $CO_2$ emissions (Burley and Katz, 2015). The magnitude of the lag depends on a number of factors, including the plate spreading rate and the rate of sea-level change, so it is not a simple task to quantify the time-dependent net effect of terrestrial and marine volcanic processes on atmospheric $CO_2$. But tantalizingly, Huybers and Langmuir (2017) have been able to reproduce ~100 kyr

glacial cycles by assuming a lag of 50 kyr and considering feedbacks between deglaciation, sea-level rise, volcanism, and $CO_2$ emissions, allowing for the temperature dependence of $CO_2$ degassing from the ocean. The $CO_2$ perturbations described here are triggered by GIA processes, and there is clear scope for further exploration of the feedbacks between the spatially-variable isostatic response to surface load changes (including the response to changes in sediment loading (Sternai et al., 2016)),

volcanism, climate change, and ice dynamics.

### 4.3 Low viscosity regions

A number of glaciated regions - Alaska, Iceland, the northern Antarctic Peninsula, and Patagonia – are situated on active or recently-active plate boundaries. The high mantle temperatures associated with such tectonic settings mean that the viscosity of the upper mantle is likely to be very low in these regions, typically $\leq 10^{19}$ Pa s (e.g. Sato et al., 2011; Auriac et al., 2013;

Nield et al., 2014; Lange et al., 2014), and hence the relaxation time of the mantle will be short, on the order of years to decades. This has two important implications: (i) the viscous response to any surface mass change prior to a few thousand years ago will have decayed, and instead (ii) viscous deformation will be dominated by the response to recent, or even contemporary, ice mass change.

Glaciated low viscosity regions are exciting places to study GIA, but the issues listed above complicate attempts to separate the geodetic signal into a response to past and contemporary change. It is typically assumed that GIA is a linear background signal that reflects the viscous response to long-past ice-sheet change and any departure from the linear trend reflects the elastic response to contemporary ice-sheet change. However, in a low viscosity region the short relaxation time of the mantle means that the solid Earth response to historical ice mass change, i.e. the GIA signal, may not be linear over the epoch of geodetic

measurements, making it difficult to isolate. Furthermore, the geodetic response to contemporary ice mass change may contain both an elastic and a viscous signal (Nield et al., 2014). The problem becomes more tractable if the viscosity of the mantle can be absolutely determined, for example, via careful analysis of GPS time series in response to known surface mass change, permitting more reliable predictions of the time-varying elastic and viscous components of deformation.

The low viscosity values inferred from GIA for regions such as Iceland and Alaska are similar to the values determined in studies of post-seismic deformation (Arnadottir et al., 2005; Johnson et al., 2009), and it has been suggested that power-law flow may sufficiently describe both processes due to the similarity of the timescales over which they take place (James et al., 2009). There is the additional complication that afterslip must be accounted for in post-seismic studies (Ingleby and Wright, 2017), but in general, the changing deformation rates observed during an earthquake cycle suggest that the Earth either follows

a power-law rheology (Freed and Burgmann, 2004; Freed et al., 2006), or a rheology comprising several different relaxation times (Pollitz, 2005; Hetland and Hager, 2006). Is there a single rheological law that can explain GIA, post-seismic deformation, intra-plate deformation, and deformation in response to sediment or lake loading (Gilbert, 1890; Dickinson et al., 2016)? It is clear that the Earth behaves differently over different timescales (Burgmann and Dresen, 2008; Watts et al., 2013),

and it will be interesting to see how parallel fields of research progress towards quantifying the rheological structure of the solid Earth, and the degree to which the rheology reflects the forcing that is applied.

## 5 Conclusions

The ideas developed by Jamieson (1865), Croll (1875), Woodward (1888), Nansen (1921), and Daly (1925) laid the foundations for the development of the sea-level equation (Farrell and Clark, 1976) and the study of glacial isostatic adjustment (GIA). With the rapid development of cross-disciplinary science in the last two decades the field of GIA has expanded beyond geodynamics, incorporating important developments from geodesy, glaciology, and seismology, whilst embracing new results from the fields of geology and geomorphology. Likely areas of future research are summarized below:

i) GIA modelling is a tool that can be used to reconstruct past ice-sheet change, but there remain large uncertainties on past global and regional ice volumes. Progress can be made by incorporating modelling elements from other disciplines, such as ice-sheet modelling, and assimilating new constraints on past sea-level change and ice extent.

ii) GIA modelling can also be used to interrogate the rheological properties of the solid Earth. Recent work has focused on understanding the role of lateral variations in viscosity and the potential for power-law rheology to explain observations of solid Earth deformation, but computational challenges remain. Work is ongoing to quantify the properties of the lower mantle – important for global-scale processes – and understand the scale at which spatial variations in Earth structure must be resolved to accurately model the GIA process.

iii) Due to the feedbacks that exist between Earth deformation, sea-level change, and ice dynamics, important insight can be gained from coupled models that consider the dynamic evolution of the global ice sheets as well as traditional GIA-related processes. Such models should ultimately include spatial variations in Earth rheology as feedbacks between solid Earth deformation and ice dynamics will be strongest in regions with low viscosity upper mantle.

iv) Novel data sets are needed to better constrain GIA – both in terms of the spatial and temporal coverage of the data, but also the type of data that can be used to provide information on GIA-related processes. Uncertainties on all data should be quantified, allowing better quantification of the uncertainties associated with GIA model predictions.

v) Comparisons between model output and data are a central component of many GIA studies, but care must be taken to account for non-GIA factors that may bias our interpretation of the data. One way to address this issue is to incorporate GIA into Earth system models and models that seek to interpret the geomorphological signature of processes such as erosion, sedimentation, mantle convection, and volcanism.

vi) Observations that are used to quantify present-day ice-sheet and sea-level change, such as tide gauge, GPS, and GRACE data, will be overprinted with a signal associated with the ongoing response of the solid Earth to past surface

load change. Quantifying the 'GIA correction' that should be applied to such data sets continues to be of vital importance as we seek to understand the processes responsible for current and future ice-sheet and sea-level change.

**Author Contribution**

PLW wrote the manuscript. Grace Nield is acknowledged below for her assistance in drafting the figures.

**Competing interests**

The author declares that they have no conflict of interest.

**Acknowledgements**

Grace Nield is warmly acknowledged for her help in producing the figures and for providing invaluable feedback during the writing of the manuscript. Holger Steffen and an anonymous reviewer are also thanked for their insightful comments during the review process. Many of the ideas explored in the final few sections of the manuscript reflect discussions between the author and researchers from a wide variety of disciplines, as well as a community effort at the 2017 IAG/SCAR-SERCE GIA Workshop to identify current research questions in the field of GIA (see https://www.scar.org/science/serce/news/ for a report
on this workshop). PLW is supported by a UK Natural Environment Research Council (NERC) Independent Research Fellowship (NE/K009958/1). This article is a contribution to the SCAR SERCE program.

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

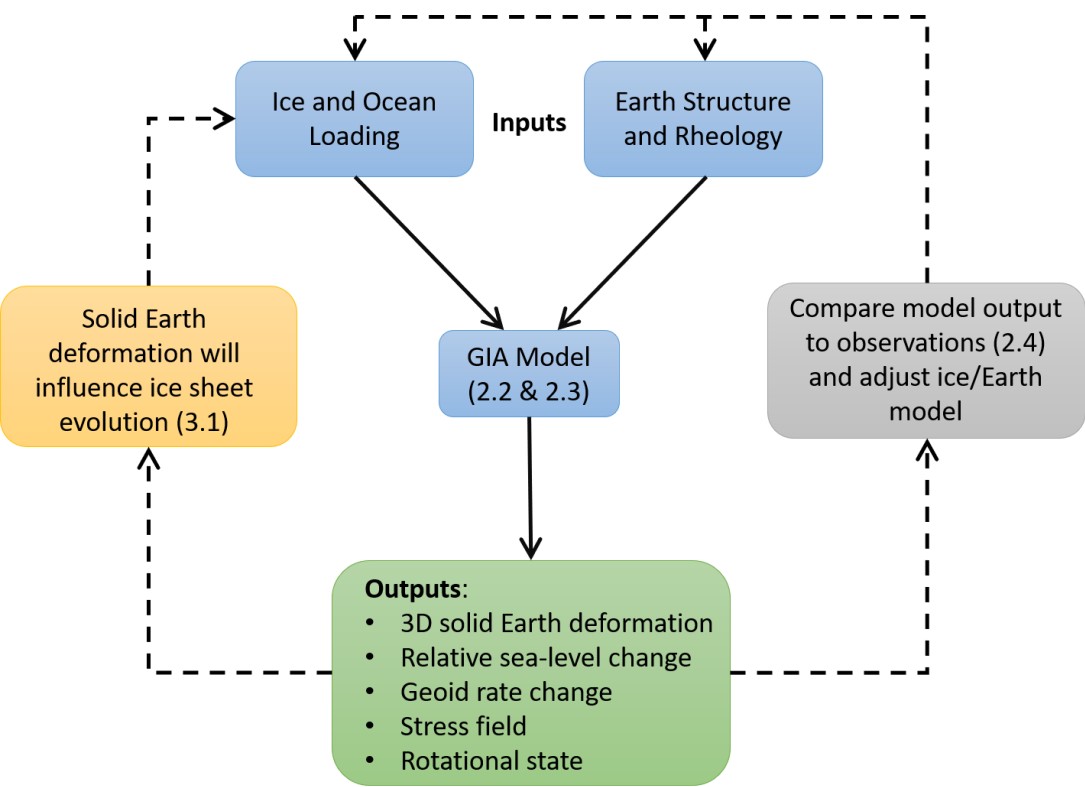


**Figure 1: Components of a GIA model.** Surface loading by ice sheets and the ocean, together with Earth properties, govern how the solid Earth deforms. Changes to the gravity field (determined by solving the sea-level equation) define how meltwater is redistributed across the oceans. Comparing model outputs to observations allows model inputs to be adjusted to achieve a better fit. Solid Earth deformation will affect ice-sheet evolution; this can be modelled with a coupled model (Sect. 3.1). Numbers refer to relevant sections in the text.


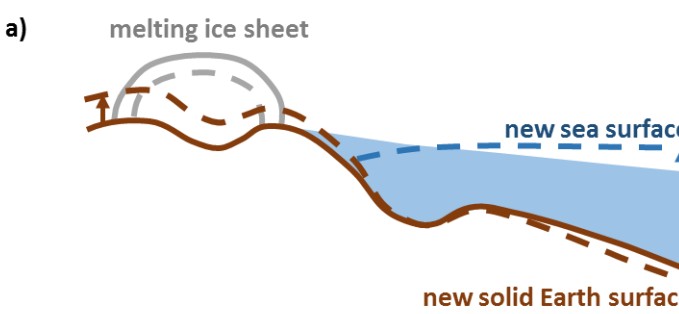

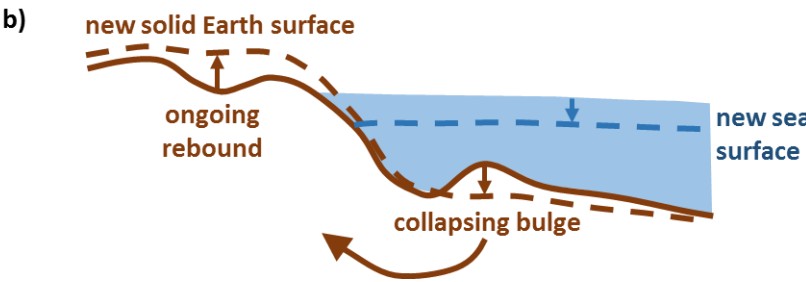

**Figure 2: Solid Earth deformation and sea-level change.** a) Ice sheet losing mass results in solid Earth rebound and a decrease in sea surface height due to the decreased gravitational attraction of the ice sheet. Both processes cause near-field relative sea-level fall. Relative sea level rises in the far field due to the addition of meltwater to the ocean. b) Ongoing solid Earth relaxation after disappearance of the ice sheet. Ocean syphoning is the process whereby peripheral bulge subsidence increases the capacity of the ocean; the result is a fall in mean sea surface height. Solid lines indicate original positions, dashed lines indicate new positions. Figure adapted from Conrad (2013).


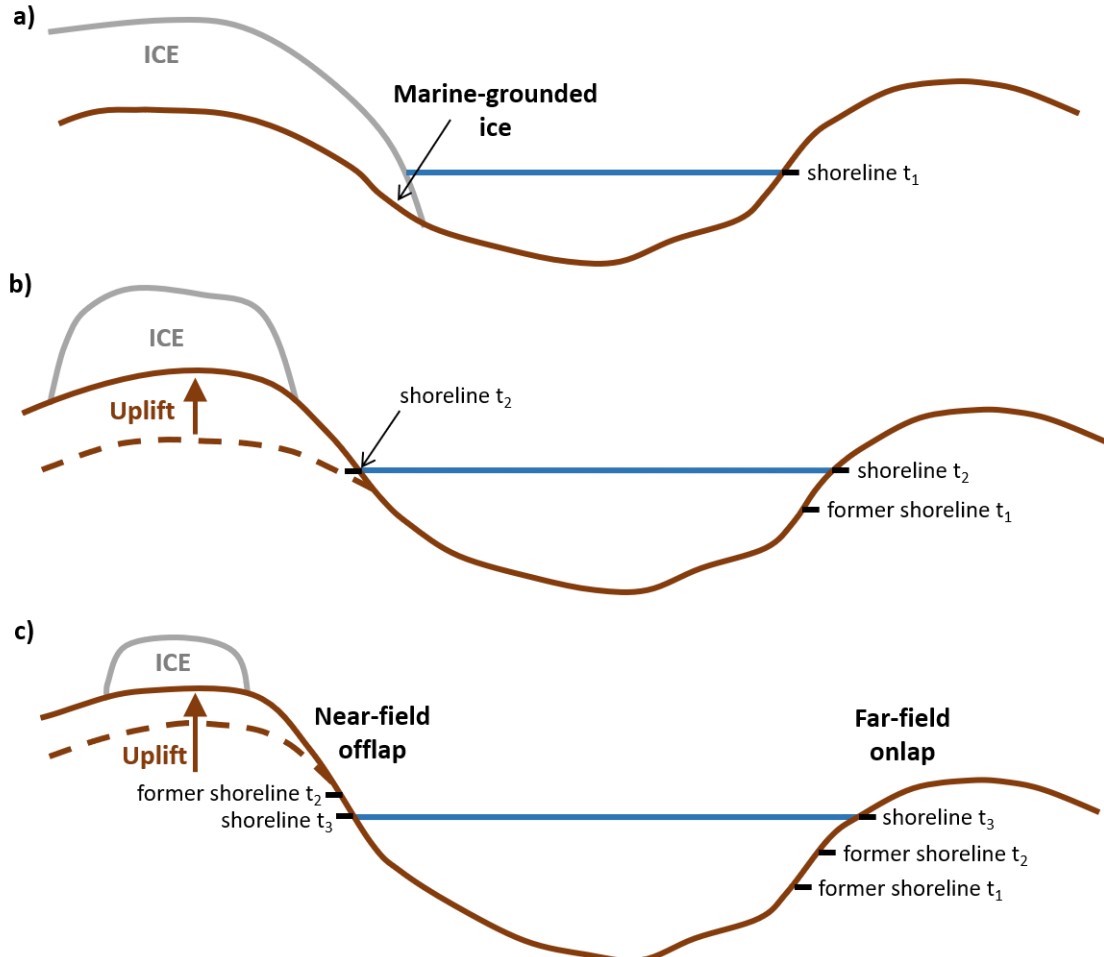

**Figure 3: Variations in ocean area.** a-b) Retreat of marine-grounded ice increases the area of the ocean over which water can be
redistributed. b-c) Onlap and offlap changes the areal extent of the ocean. In the near field of a melting ice sheet, rebound results in local
sea-level fall, causing the shoreline to migrate offshore (offlap). In the far field of a melting ice sheet, sea-level rise causes the shoreline to
migrate onshore (onlap). $t_1$, $t_2$, $t_3$ refer to the times represented in a) b) and c). Figure adapted from Farrell and Clark (1976).


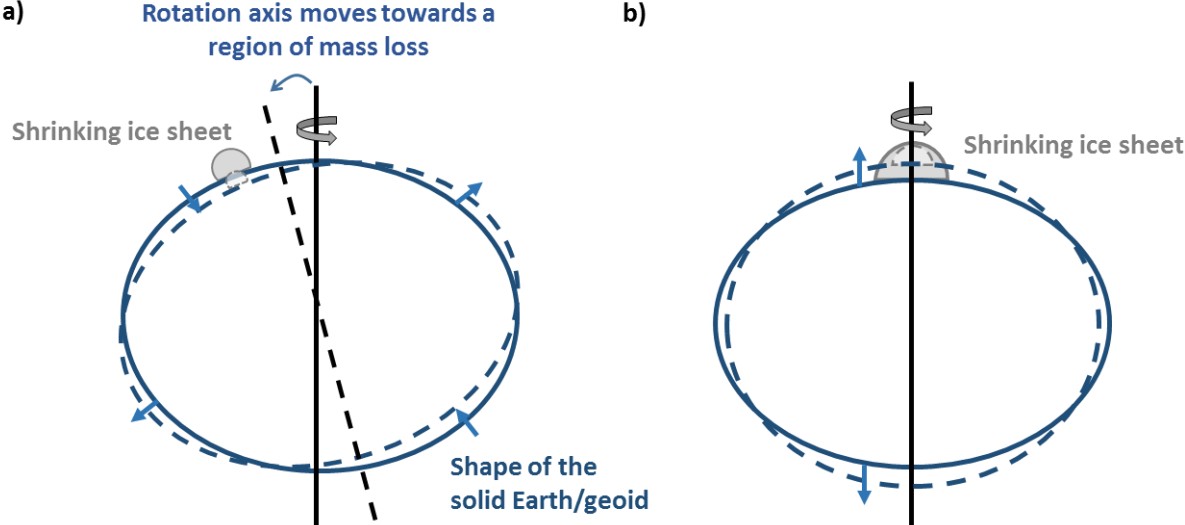

**Figure 4: Rotational feedback.** a) Earth's rotation vector moves towards a region of mass loss, causing a change in the shape of the solid Earth and the geoid. Relative sea level rises and falls in opposing quadrants of the Earth. b) Polar ice loss results in a decrease in the oblateness of the Earth ($\dot{J}_2$). Solid lines indicate original positions, dashed lines indicate new positions.

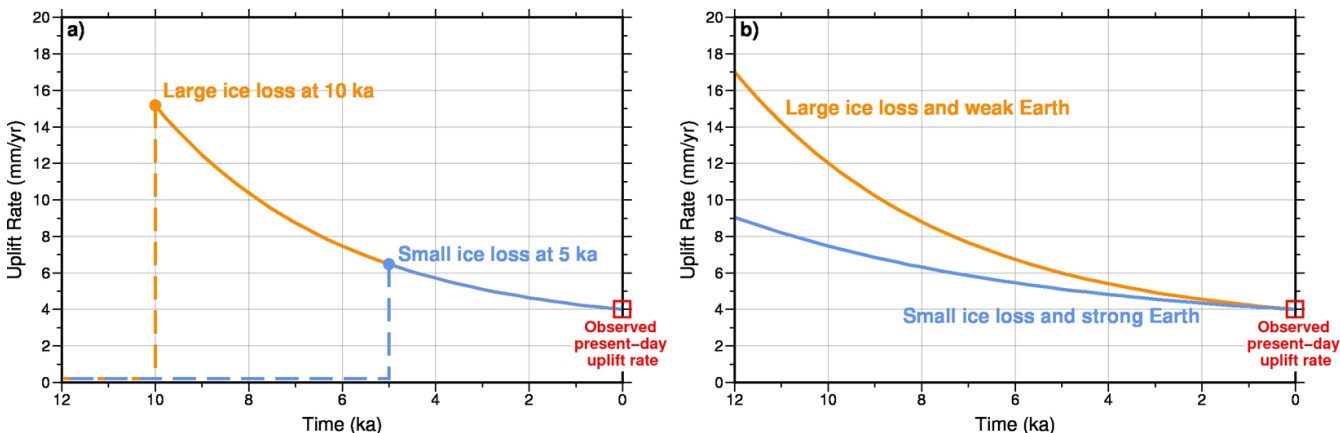

**Figure 5: Non-uniqueness in GIA modelling.** a) Trade-off between the timing and the magnitude of past surface load change: large ice loss at 10 ka can result in the same present-day uplift rate as smaller ice loss at 5 ka. b) Trade-off between ice load history and Earth rheology: large ice loss combined with a weak rheology can produce the same present-day uplift rate as small ice loss combined with a strong rheology.

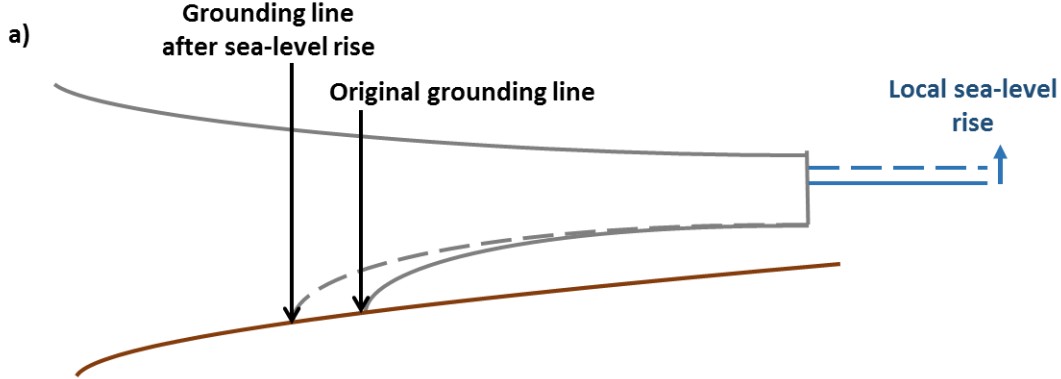

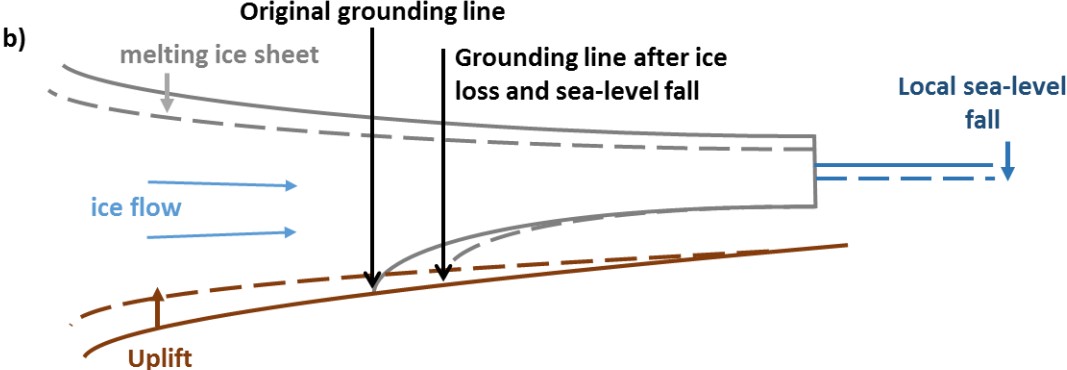

**Figure 6: GIA-ice dynamic feedbacks.** a) Far-field ice melt leads to local sea-level rise which causes a retreat in the position of the grounding line (the point at which an ice sheet starts to float). b) Near-field ice melt leads to solid Earth rebound which, combined with a decrease in gravitational attraction, results in local sea-level fall. This has a stabilizing effect on the ice sheet and results in grounding line advance. Solid lines indicate original positions, dashed lines indicate new positions.

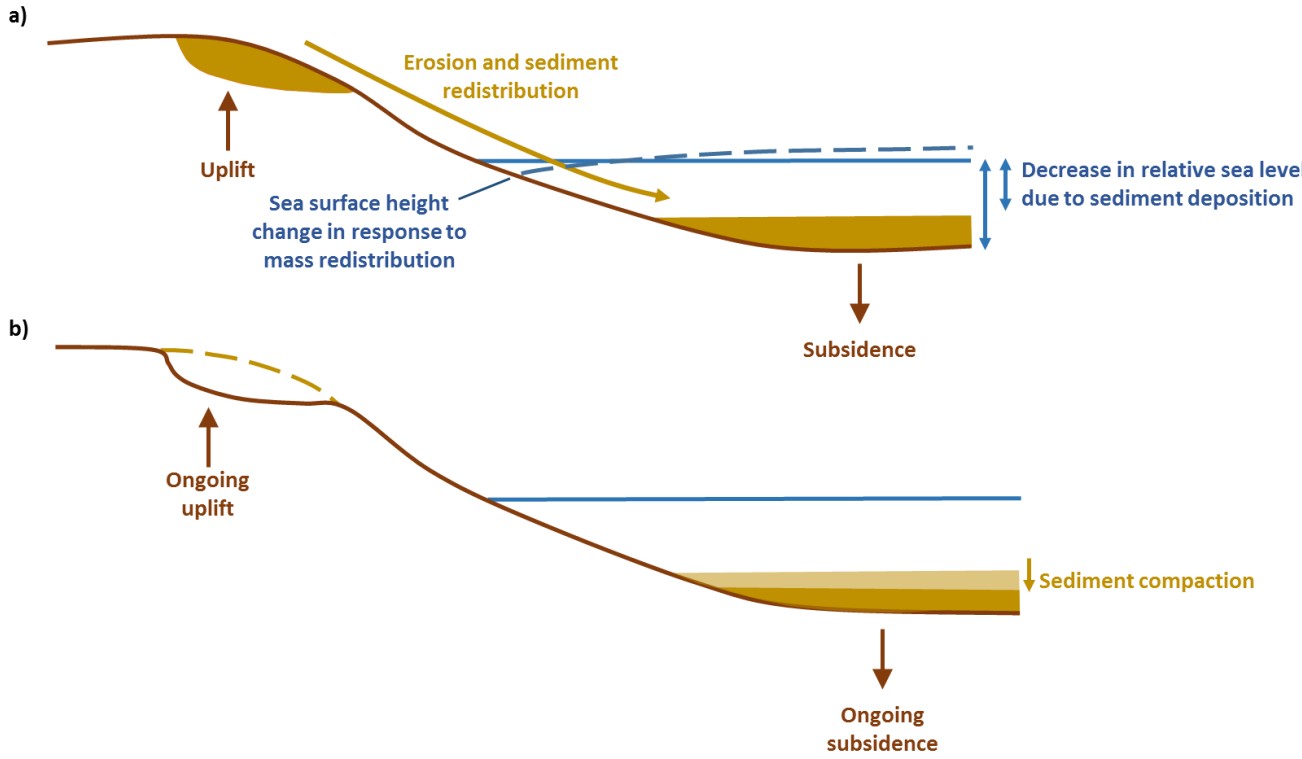


**Figure 7: Effect of sediment redistribution on GIA.** a) Sediment erosion and deposition results in solid Earth deformation and a local reduction in water depth. The net redistribution of mass (solid Earth and sediment) will also change the shape of the geoid, and hence local sea level. b) Sediment compaction over time results in a small increase in local water depth. Solid lines indicate original positions, dashed lines indicate new positions.