# Peer review of "Glacial Isostatic Adjustment modelling: historical perspectives, recent advances, and future directions"

_Earth Surface Dynamics, 2018_

## Referee Comment (RC1) · Anonymous Referee #1 · 28 Feb 2018

Review of *"Glacial Isostatic Adjustment modelling: historical perspectives, recent advances, and future directions"* by Pippa L. Whitehouse, submitted to: Earth Surf. Dynam. Discuss.

**1. General comments.**

This is a comprehensive review of the Glacial Isostatic Adjustment (GIA) problem, in which the emphasis is about the historic development, the recent advances and possible future directions. Despite the word 'modelling' appears in the title, there is little about the technical aspects of GIA and as a consequence the approach is rather qualitative and focussed on the description of the geophysical processes involved in GIA. A series of nice illustrations are proposed, useful to understand how GIA operates on global and regional scales, something that is certainly of interest for beginners in this field. The strength of the paper is in its completeness and organisation, which make it very easy to follow despite its considerable length. The weakness is in some missing details and references and sometimes in a too involved description of the various topics. In some parts, the paper seems to have been written in a rush, so that some smoothing and re-thinking is recommended before it can be considered for publication.

I have a series of minor to moderate remarks, various considerations, notes and hints, listed in the following, which I hope are useful to improve an already very good contribution.

**2. Specific comments**

Line 1ff. In some previous works (e.g., DOI:10.1007/s10712-016-9379-x), a distinction is made between the Earth's response to past ice melting and that to present ice melting. Of course the physical principles are the same, but a few explicatory words can be of help. For example, when dealing with low-viscosity regions in Section 4.3, it should be clear that in this case we are dealing with the GIA caused by recent melting.

L6. Not only to 'global' ice sheets; in some parts GIA is in response to regional ice sheets (e.g., again Section 4.3).

L7. Actually, GIA 'follows' the entire history of glaciations, which is 100s of kyrs long. I do not think GIA should only be viewed only as the response to glacial unloading. Perhaps some words to distinguish between 'post-glacial rebound' and GIA can help, in this respect.

L18. (LGM, ~21,000 years ago).

L44. GIA can be "seen" (and it was actually seen) by looking at the migration of the shorelines in northern Europe, as summarised here. However, in the whole paper there is not so much attention about the general effects of GIA on the time evolution of the past Earth's topography (e.g. former "land bridges" are only mentioned at L759). May be it could be useful to look at the literature of contemporary GIA investigators, in this respect. I am sure

that a little section about this important aspect of GIA modelling would add some value to this review.

L52ff. I suggest to change item (viii) into "interpreting the gravity field of the Earth and its rotational state" or something like that.

L60ff. This is a remarkable historical overview, indeed. I have some miscellaneous observations. *i)* Did Jamieson only consider elastic deformation? *ii)* Was his work influenced by the ideas of Airy about isostasy? *iii)* It is probably important to mention, for later comparisons, the famous Haskell value for mantle viscosity. *iv)* Regarding uniform viscosity Earth models, the Darwin spherical model has had a role in the development of GIA models (see the review of Peltier, 1974, who should be quoted e.g., at L176 between O'Connell and Chatles, in my opinion). *v)* I am surprised to see no mention to the fundamental work of Love, who with his "Love numbers" certainly prepared the advent of the modern GIA models, starting from Farrell and Clark (1976).

L175. I think 'is the equipotential' is better than 'is an equipotential'.

L176. This can be shortened into 'in the absence of winds and currents'.

L180ff. This is nice illustration, in words, of the concept behind the 'k' Love number.

L187. I would suggest 'constrained' *in lieu* of 'determined'.

L190ff. The role of the sea level equation (SLE) in GIA modelling has been recently reviewed in detail in DOI:10.1007/s10712-016-9379-x.

L190ff. Eq. (1) and the material that follows is fairly good, but it can be improved, I have a suggestion. Why not starting with the SLE in the native form $S=N-U$ (new equation 1)? This would be helpful, since in this way one can define relative sea level change ($S$) and absolute sea level change ($N$), and vertical displacement ($U$) since the onset. These are quantities of fundamental importance for a full understanding of the remainder of the paper. If this is agreed, Eq. (1) becomes Eq. (2) and so forth… Also, the unnecessarily cumbersome symbol DeltaSL could simply become $S$ and so on al L196 and L201 (and in other places, I think).

L194. Given that the unknown in the sea-level equation is $S$ (provided that the history of the changes in ice thickness $I$ is assumed to be known), one may wonder how surface displacements $U$ or horizontal displacements $V$ are obtained for e.g. comparison with GNSS data, discussed later in the paper.

L196: … is the change in *relative* sea level…

L201. This can be rearranged into: … are convolutions in time (over the ice sheets history) and in space (over the surface of the ice sheets and of the oceans….).

L204. It is not actually a uniform shift 'in the geoid', it is a uniform shift in *relative* sea level, according to the equation labelled by (1) in the present form of the paper.

L209. The definition of eustatic sea level is important in GIA modelling, but here it is only mentioned *en passant*. Seen the beautiful historical Section 2.1, some words could be certainly spent on the work of Edward Suess on the concept of eustasy in his book *"La Face de la Terre"*. To be completely fair (and admittedly pedantic), eustasy is a static concept, only dependent on the amount of melt water change, on the density of water and on the area of the oceans surface. However, in his Eq. (2), the author has accounted for variations of the area of the oceans by a time-dependent $A\_o(t)$. This is a dynamic effect, however, since it also depends e.g. on Earth's rheology. So, I suggest to warn the reader about this caveat, or to re-write the SLE using an effectively constant $A\_o$. I would personally prefer the first solution. I also note that in the recent literature the term 'eustatic' has been substituted by '*barystatic*' following the work of Gregory et al. (2013), something that could be mentioned or not. I finally observe that when a time-dependent $A\_o(t)$ is accounted for (this is the case for migrating shorelines), the sea level equation becomes non-linear; otherwise it is linear for an effectively constant $A\_o$. This an important qualitative difference that should be mentioned, in my opinion.

L212. They are not really 'perturbations'; maybe they are 'terms'.

L222. The Maxwell rheology is describing a fluid behaviour, not a solid behaviour. In fact, in a creep experiment, just immediately after the instantaneous elastic response, the dashpot of the Maxwell body works as a Newtonian viscous fluid. The same at L223, and in other places like L570, for instance.

L226. If a power law is assumed, the Love numbers formalism is not viable since the problem becomes non-linear.

L234. The lithosphere is far from being purely elastic (see e.g. Ranalli, *The Rheology of the Earth,* or DOI: 10.1029/RG021i006p01458, or the papers by Burov, e.g., DOI: 10.1029/94JB02770). The main point is "why the elastic lithosphere approximation is so common in GIA studies"? (despite the evidence for a complex rheological profile).

L250. What 'self consistent' means here?

L247ff. All these extensions of the sea level equation make it a non-linear equation; this is an important point. See also my comment to L209 above.

L256ff. It can be worth to say that the effects of Earth rotation introduce a very long wavelength pattern mostly characterised by harmonic degree l=2 and order m= +/-1 terms. This high-energy component of sea-level change is clearly visible in the form of large lobes in the maps GIA fingerprints, see e.g, http://dx.doi.org/10.1016/j.gloplacha.2016.05.006.

L256ff. I would rephrase as follows "Since GIA alters the mass distribution of the Earth, it changes its (off-diagonal) moments of inertia, which in turn perturbs…" or similar.

L261. Why 'over longer time scale'? Actually, rotationally induced Earth deformations also occur on short time scales, even elastically (i.e., instantaneously).

L262. Again, I do not capture the rationale for separating short from long time scales.

L274. I would avoid the term 'transient', since this term can also refer to non steady-state (e.g., Burgers) rheological effects. I think that 'time-dependent' could be a possible alternative.

L285. I am not sure that the meltwater fingerprints can be immediately associated to (or 'based on') the theory of Woodward. Indeed, Woodward used a rigid Earth and ignored the oceans self-attraction (only accounting for the gravitational attraction between the point ice load and the ocean mass). Hence, his fingerprints are approximations of the 'modern ones' based on a more realistic modelling.

L285 and L289. The term 'fingerprint' in the GIA context has been coined by Plag and Jüettner (2001) and adopted in numerous studies since then (see Plag HP, Juettner HU Inversion of global tide gauge data for present-day ice load changes - scientific paper - Mem Natl Inst Polar Res 54:301 special issue, 2001), and this should be fully acknowledged.

L293ff. One may simply wonder *why* peripheral bulges regions exist!

L297. If my hints at L190ff above are followed, the reader will be greatly facilitated here, where the concept of absolute sea level is utilised. Why 'mean'? I do not think it is necessary.

L300 and 301. I am afraid I do not understand this couple of lines. Global (absolute) sea-level change obtained from altimetry is normally corrected for the effects of GIA, so ocean syphoning and all the other processes involved are certainly taken into account. I think the paper of Tamisiea (2011) is dealing with other aspects of GIA. It is possible, however, that I am missing the point, here.

L305. The 'continental levering' effect should be mostly visible far from the polar regions, right?

L312ff. Perhaps, it can be of interest to note that before the advent of the so-called pseudo-spectral method, a fully spectral approach was utilised (see e.g., the paper by Plag and Juettner quoted above).  Regarding the methods in general, it could be instructive to explain *why* certain methods are used instead of others. For instance, the pseudo-spectral method is now standard when dealing with spherically symmetric Earth models with linear rheology. Similarly, the finite-element approach of Wu and coworkers is motivated by the

introduction of a power-law rheology (for which the superposition principle does not hold), and so on...

L320. Here and in other places, the word 'self-consistent' should be used in a more specific way. I think that in the context of GIA the term 'gravitationally self-consistent' has been introduced to say e.g. that the change in the shape of the oceans determined by solving the sea level equation is consistent with the gravitational field (the oceans surface is an equipotential ad mass conservation is ensured). Similarly, 'topographically self consistent' indicates that the solution of the sea level equation for a variable topography is consistent with the present day topography (and with the gravity field). See the works of WR Peltier, where (I think) this terminology has been introduced first. See also point L250.

L321. I do not think that *all* integral equations need to be solved by iteration. Iteration is often invoked to solve non-homogeneous Fredholm equation of the second kind, which is the type of equation the sea level equation is. In any case, most importantly, the physical reason for which iterations are needed is that the change on the ocean mass distribution is not known a priori, contrary to the ice distribution.

L322. The Section on Data is very smooth. Regarding 2.2.3, I only observe that before GPS data, very long baseline radio-interferometry (VLBI) data have also been employed to test GIA models and to constrain the Earth's viscosity profile. This can be traced easily in the literature.

L400ff. I do not fully agree on the role of horizontal GPS observations; their recognised sensitivity to the shallow upper mantle rheology and to the presence of lateral variations in the Earth's mantle properties should be regarded as an advantage, not as a limitation.

L402ff. Similarly, I do not think GIA model predictions are typically provided on a reference frame whose origin lies at the centre of mass of the solid Earth. Quite often, instead, they are given in the reference frame of the center of mass of the whole Earth system, so that they can be directly compared to geodetic information (actually, in GIA modelling the transformation between the two frames is straightforward). I agree on the reference frame origin uncertainties, which is indeed a major problem.

L416ff. This is true, but I think the main problem with GRACE is that it observes gravity field variations due to all surface and internal sources (including e.g. mantle dynamics associated with non-GIA processes, post-seismic deformations, and so on). I also note that some very useful insight into the actual meaning of GRACE data has been recently given by BF Chao in DOI:10.1007/s00190-016-0912-y.

L444ff. I am not really getting the point here. Can examples and/or citations be given to support or explain this sentence?

L454. As far as I know, a possible interaction between GIA and seismicity has been first proposed by Gutenberg and Richter in their book *Seismicity of the Earth and Associated*

*Phenomena* (1949), and there is a long story of relevant contributions to this field since the 90s, probably listed in the bibliography of Dr. Steffen's works. Some considerations along these lines could help to put flesh on the bones of this succinct subsection.

L467. I agree that the definition of intervals for the viscosity of the upper and lower mantle has been (and is) a key-result in GIA modelling. But I also think that the recognition that GIA data requires a rheologically layered mantle (i.e. a viscosity jump between the upper and a the lower mantle) is even more important. It would be very useful to trace this key-result in the literature.

L475ff. Although the physical process is the same, it is important to let the readers know that contrary to the studies mentioned at ~L470, in those quoted here the source of GIA is recent ice melting from small size glaciers and ice caps. By the way, I definitively agree that defining these low-viscosity layers is a key result of GIA modelling.

L479-L491. This paragraph is OK, but still it is difficult to see sharp key results. Itemisation can help? May be the text can be modified accordingly. In any case, since the 70s the Toronto University GIA school has given important contributions into the definition of the ice volume change since the LGM with the help of GIA modelling, and this should be fully recognised, in my opinion.

L492ff. I realise that summarising all the attempts to constrain the configuration of individual ice sheets in a short paragraph is not easy. But here one is left with the impression that all the studies mentioned converge to similar results. However this is not the case; for instance, as the author of this manuscript knows, different studies show diverging results for the history of melting chronology of the Antarctic ice sheet during the late Holocene. Actually, resolving this uncertainty could constitute one of the future challenges in GIA modelling for Section 4 below.

L501. The sentence should be probably tuned differently for the Greenland and Antarctic Ice Sheets. In any case, GIA corrections are particularly uncertain because of the uncertainties about the melting history of Antarctica.

L522ff. Just a comment. There are uncertainties associated with limited data availability and modelling capability. However, I think that the uncertainties described by Tamisiea (2011) are of different nature, resulting from genuine misunderstanding of the physical meaning of the various terms of the sea-level equation.

L584. Yes, this is an important question. Concerning global GIA models, I do not think we are now in the position of saying that 3D non-Newtonian models perform better that 1D Maxwell models in explaining e.g. the sea-level variations observed during the Holocene. So, I do not know whether an increased model complexity is indeed required.

L615. I am not really an expert in sedimentary isostasy, but I know that sometimes sediment loading and sediment compaction occur at a very small scale. In view of that, is there some

indication of the spatial resolution of the GIA models in which these effects are taken into account? Maybe some information could be conveyed to the readers in this respect. I am also curious about how mass is conserved in these sedimentary GIA models, since this would require detailed information about the sediments (re)distribution, etc…

L653. The total mass change should be zero, by mass conservation; probably 'mass redistribution' is meant here? A few lines below (L669): I am missing why an inverse approach is neither dependent on the ice loading history nor from the Earth structure. What does it depend upon? Overall, this whole Section 3.4 is definitively not written for newbies and is somewhat confusing, in my opinion.

L676. Section 4: future directions. This is a very nice (and very personal) view of the possible future developments of GIA modelling. I do not have specific comments on that.

L827. A very short conclusion that sounds a bit vague. I would have preferred a few (better if itemised) statements summarising three or four take home messages.

Hope this helps.

---

## Referee Comment (RC2) · H. Steffen (Referee) · 28 Feb 2018

Review papers can be a dangerous task to undertake, and can also be challenging to comment on. Arguments of different referees can be condratictory: too long, too short, too many references, too few, not all is covered, too much is covered, too much focus on one issue, chapters order should be the other way around and so on and so forth. As an author, it is challenging to satisfy all those views and partly needs a thick skin, also once it is published and more people comment on it.

From my perspective, Pippa Whitehouse reviews glacial isostatic adjustment (GIA) modelling very well and I think that this paper will be highly regarded and cited, es-

pecially as it draws a bow from the historical development over recent advances to possible future directions posing a few interesting research questions to take on for the wider community.

The paper is in most parts well written and additionally supported by a couple of figures that help explain major GIA-related processes. Much thought was given on recent and future developments and may likely reflect many personal views, I note though many discussions that Pippa does on conferences, thus I think she has a good overview of ongoing and future works.

I recall the GIA workshop in Reykjavik in September 2017 where the 10 top research questions in GIA research were collected. I guess some of them also arise while reading this paper and perhaps it is an option to pick that up in the Conclusions pointing to the SCAR-SERCE website.

There also are a few paragraphs/sections that are very brief with a few common references while others are well developed with a large number of references. I guess this is due to the personal research interest of the author, and I try to give some help below so that brief parts can be extended.

I hope that my comments in the following are taken in the positive manner with which they are intended. I apologize right away that some comments and mentioned references are due to my "Fennoscandia-biased" eye, so please take them as suggestion only, especially when I write "I'd suggest"...

General remarks

L12: I would not agree with "in detail" when you mean "field of GIA" - there is so much more to discuss - but I would support it when you say "field of GIA modelling".

First paragraph of Introduction: I'd suggest to add 1-2 sentences about the term "postglacial rebound" which you mention three times in this paper, e.g. L780 "postglacial rebound is not the only GIA-related process". It would be good for the reader to learn

about this term and how it relates to GIA, especially as it is still often used synonymously in publications.

L44-59: as your paper has next to modelling a strong bias to sea level, it is perhaps good to mention a couple of other review papers the reader can look at, e.g. Whitehouse (2009, I really missed that in the references), Steffen & Wu (2011), Milne (2015), Spada (2017)

L64/65: The sentence is per se ok but I note that Celsius was not the first to cut marks for the sea level in rocks, see Mörner (1979). Celsius' intention though was to allow rigorous measurements, especially for future generations. See Martin Ekman's new book (2016) about Celsius, section 5.4 therein.

Section 2.2: This section discusses only GIA models applying the sea-level equation but misses among others flat Finite Element models. Also GIA-fault models are not yet connected to it. I'd suggest to either alter the title slightly including the SLE or add another short section with a discussion of models not involving the SLE, see my comments on L216-218.

L183/184: you can well mention here that the "system Earth" has further contribution that you will pick up later.

L209: I'd suggest to add references Lambeck et al. (2001) and Milne et al. (2002) here and add a few words on the issue briefly discussed in both papers, that is that inconsistent use of this term has caused some confusion earlier, thus one should be aware that some publications may use it in a wrong way.

L216-218: Although this paragraph intends to make a link to following sections, I became irritated while reading the next sections. I expected a discussion of the different quantities of the SLE, especially as you began in 2.2.2 with the solid Earth, thus I expected the ice thickness evolution next and so on. Putting 2.2.2 aside (and perhaps move it somewhere else) Section 2.2 would deal with the SLE only because 2.2.3-2.2.5

deal with SLE but not with GIA models as indicated in the title of 2.2. So a suggestion could be to rename section 2.2 to "Development of the sea-level equation" or something similar, section 2.2.1 to "The original form of the sea-level equation", moving 2.2.2 after 2.2.4 to have old 2.2.1, 2.2.3 and 2.2.4 as new 2.2.1, 2.2.2 and 2.2.3. Then put old 2.2.2 together with 2.2.5 and perhaps other GIA models without application of the SLE including GIA-faults models (R. Steffen et al. 2014a) to a longer new "Numerical methods for modelling GIA" section with old 2.2.2, missing GIA without SLE, old 2.2.5 to new 2.3.1, 2.3.2 and 2.3.3. This should also fit the title of section 2.

Section 2.2.5: I'd suggest if possible add a few words on the temporal and spatial resolution of the models and how it has changed over the years. Anything easy to say about the run time for such models - really general in terms of minutes, hours or days?

Section 2.3: (1.) You briefly mention optimal location later but I'd suggest to place a few words already here. I note that there is not only the study by Wu et al. (2010) on optimal GPS, later on Steffen et al. (2012, 2014) also discussed optimal gravity and RSL data, respectively. (2.) You miss a quite interesting, though also difficult to analyse data set: tide gauges - just mentioned two times in the manuscript. However, it is a crucial data set in all sea-level rise discussions. I'd suggest to add a section here.

Section 2.3.3: I miss the studies by Nerem & Mitchum (2002) and Kuo et al. (2004) using SAR together with tide gauges.

L396-398: I'd suggest to mention the GIA-frame approach introduced in Kierulf et al. (2014) that deals with this problem and can be regarded quite successful when checking the discussion therein.

L398-401: Getting information on the subsurface structure from horizontal velocities was the aim of Milne et al. (2004), Steffen et al. (2007) and Steffen and Wu (2014). The latter two papers are quite technical and I admit rather complex for reading but they support your conclusion that horizontal rates have not yet been used to their full potential. However, studies exist.

L416: suggest to add "e.g." to the references

L424-426: in view of the relationship between uplift and gravity change, the study by Olsson et al. (2015) should be mentioned.

L439: strongly suggest to change "may not be" to "are not"

L449/450: Argus et al. (2014) should be mentioned along the three papers.

Section 2.3.6: one of the rather short sections where much research has been undertaken, see the works by Wu, Johnston, Klemann, Kaufmann, Lund, R. Steffen etc. This section should be expanded. Stewart et al. (2000) is certainly a key paper but important studies were already published in the late 70s and much research was triggered by the studies of Spada et al. (1991), Wu & Hasegawa (1996a,b) and Arvidsson (1996), to name a few.

L455/456. The unloading is not only able to trigger postglacial faulting but also more recent (historic) earthquakes can be linked to GIA, see Brandes et al. (2015). Since a few years the term "postglacial fault" or "postglacial faulting" is thus under discussion, see e.g. Lund (2015). My personal preference is "glacially induced fault".

L458: triggering slip on pre-existing faults was a major result of R. Steffen et al. (2014b).

L495: I'd suggest to add Nordman et al. (2015) to Fennoscandia (Lev Tarasov's North-European model), and Lambeck (1995) for the British ice sheet. What about Patagonia and the work of Ivins and James (2004)?

L498: Lambeck et al. (e.g. Lambeck et al. 2014) and Tarasov also work on global models and should be named.

L577: Klemann et al. (2008) should be cited for the effect of plate boundaries.

L578: Wang & Wu (2006a,b) also analyzed the effect of a 3D lithosphere (it's in the paper titles).

Section 3.3: In view of this section, L677/678, Section 4.2 and the Conclusions you can state that GIA is part of Earth System Modelling and should always be investigated in an interdisciplinary context.

L688-691: I miss a reference here.

L725: This relates to the concept of "underwater GPS". Future measurements may help solving this question and it is perhaps worth to mention that research is going on in this field (e.g. Ramesh et al. 2016, Honsho & Kido, 2017).

L729: I'd suggest stating "use of satellite gravity data". Terrestrial gravity data are not a gap-filler for GPS in view of the effort to perform a single measurement.

L777: The studies by Schmidt et al. (2013) and Kutterolf et al. (2013) should also be mentioned.

L830: "central component of any GIA model" implies that flat FE models are no GIA models..., suggest "any GIA model"="the majority of GIA models"

Figure 1: In view of the basic equations the outputs are not complete. Rotation and stress are missing, which are both quantities that you discuss in your manuscript. Should be added.

Technical corrections

L97: The first ice(-)cap appearance should be without dash

L827: correct brackets for Daly (1925)

References (not cited in the manuscript)

Arvidsson, R. (1996), Fennoscandian earthquakes: Whole crustal rupturing related to postglacial rebound, Science, 274, 744–746, doi:10.1126/science.274.5288.744.

Brandes, C., Steffen, H., Steffen R. and Wu, P., 2015. Intraplate seismicity in northern Central Europe is induced by the last glaciation. Geology 43, 611-614.

Ekman, M., 2016. The man behind "Degrees Celsius": A pioneer in investigating the Earth and its Changes. Summer Institute for Historical Geophysics, 159 pp., ISBN: 978-952-93-7732-9 (see also http://www.historicalgeophysics.ax/#nyarebok)

Honsho, C., & Kido, M.(2017). Comprehensive analysis of traveltime data collected through GPS-acoustic observation of seafloor crustal movements. Journal of Geophysical Research: Solid Earth, 122. https://doi.org/10.1002/2017JB014733

Klemann, V., Martinec, Z., Ivins, E. R. (2008): Glacial isostasy and plate motion. - Journal of Geodynamics, 46, 3, pp. 95-103. doi:10.1016/j.jog.2008.04.005

Kuo, C.Y., Shum, C.K., Braun, A., Mitrovica, J.X., 2004. Vertical crustal motion determined by satellite altimetry and tide gauge data in Fennoscandia. Geophys. Res. Lett. 31, L01608, doi:10.1029/2003GL019106.

Kutterolf, S., Jegen, M., Mitrovica, J. X., Kwasnitschka, T., Freundt, A. and Huybers, P. J. (2013) A detection of Milankovitch frequencies in global volcanic activity. Geology, 41 (2). pp. 227-230. DOI 10.1130/G33419.1.

Lambeck, K., 1995. Late Devensian and Holocene shorelines of the British Isles and North Sea from models of glacio-hydro-isostatic rebound. J. Geol. Soc. Lond. 152, 437-448.

Lambeck, K., Yokoyama, Y., Johnston, P., Purcell, A., 2001. Corrigendum to "Global ice volumes at the Last Glacial Maximum and early Lateglacial". Earth and Planetary Science Letters 190, 275.

Lund, B., 2015. Palaeoseismology of glaciated terrain. In: M. Beer et al. (eds.) Encyclopedia of Earthquake Engineering, Springer-Verlag, Berlin, Heidelberg, pp. 1765-1779, doi: 10.1007/978-3-642-36197-5_25-1.

Milne, G. 2015. Glacial Isostatic Adjustment. In: Shennan et al. (eds.) Handbook of Sea-Level Research. John Wiley & Sons. pp. 421-437, ISBN: 978-1-118-45258-5
[Figure]

Mörner, N.-A., 1979. The Fennoscandian uplift and late Cenozoic geodynamics: geological evidence. GeoJournal 3.3, 287–318, doi:10.1007/BF00177634.

Nerem, R., and G. Mitchum (2002), Estimates of vertical crustal motion derived from differences of TOPEX/POSEIDON and tide gauge sea level measurements, Geophys. Res. Lett., 29(19), 1934, doi:10.1029/2002GL015037

Maaria Nordman, Glenn Milne, Lev Tarasov. Reappraisal of the Angerman River decay time estimate and its application to determine uncertainty in Earth viscosity structure, Geophysical J. Int., 201, 811–822, doi: 10.1093/gji/ggv051, 2015.

P.-A. Olsson, G. Milne, H.-G. Scherneck, and J. Agren. The relation between gravity rate of change and vertical displacement in previously glaciated areas. Journal of Geodynamics, 83:76{84, 2015.

Ramesh, R., Jyothi, V.B.N., Vedachalam, N., Ramadass, G.A. & Atmanand, M.A. 2016, "Development and Performance Validation of a Navigation System for an Underwater Vehicle", Journal of Navigation, vol. 69, no. 5, pp. 1097-1113

Schmidt, P., B. Lund, C. Hieronymus, J. Maclennan, T. Árnadóttir, and C. Pagli (2013), Effects of present-day deglaciation in Iceland on mantle melt production rates, J. Geophys. Res. Solid Earth, 118, 3366-3379, doi:10.1002/jgrb.50273.

Spada, G. (2017). Glacial Isostatic Adjustment and Contemporary Sea Level Rise: An Overview. Surv Geophys (2017) 38:153–185

Spada, G., D. A. Yuen, R. Sabadini, and E. Boschi (1991), Lower-mantle viscosity constrained by seismicity around deglaciated regions, Nature, 351, 53–55, doi:10.1038/351053a0.

Steffen, H., Wu, P., Kaufmann, G., 2007. Sensitivity of crustal velocities in Fennoscandia to radial and lateral viscosity variations in the mantle. Earth Planet. Sci. Lett. 257 (3–4), 474–485, doi:10.1016/j.epsl.2007.03.002.

Steffen, H., Wu, P., and Wang, H.: Optimal locations for absolute gravity measurements and sensitivity of GRACE observations for constraining glacial isostatic adjustment on the northern hemisphere, Geophys. J. Int., 190, 1483–1494, doi:10.1111/j.1365-246X.2012.05563.x, 2012.

Steffen, H. and Wu, P.: The sensitivity of GNSS measurements in Fennoscandia to distinct three-dimensional upper-mantle structures, Solid Earth, 5, 557-567, https://doi.org/10.5194/se-5-557-2014, 2014.

Steffen, H., Wu, P., and Wang, H.: Optimal locations of sea-level indicators in glacial isostatic adjustment investigations, Solid Earth, 5, 511-521, doi:10.5194/se-5-511-2014, 2014.

Steffen, R., Wu, P., Steffen, H., Eaton, D.W., 2014a On the implementation of faults in finite-element glacial isostatic adjustment models. Computers & Geosciences 62, 150-159.

Steffen, R., Steffen, H., Wu, P., Eaton, D.W., 2014b. Stress and fault parameters affecting fault slip magnitude and activation time during a glacial cycle. Tectonics, 33, 1461–1476.

Wang, H., Wu, P., 2006a. Effects of lateral variations in lithospheric thickness and mantle viscosity on glacially induced relative sea levels and long wavelength gravity field in a spherical, self-gravitating Maxwell Earth. Earth Planet. Sci. Lett. 249, 368–383.

Wang, H., Wu, P., 2006b. Effects of lateral variations in lithospheric thickness and mantle viscosity on glacially induced surface motion on a spherical, selfgravitating Maxwell Earth. Earth Planet. Sci. Lett. 244, 576–589.

Whitehouse, P., 2009. Glacial Isostatic Adjustment and Sea-level Change: State of the Art Report. TR-09–11, Svensk Karnbranslehantering AB.

Wu, P., and H. S. Hasegawa (1996a), Induced stresses and fault potential in eastern

Canada due to a realistic load: A preliminary analysis, Geophys. J. Int., 127, 215–229, doi:10.1111/j.1365-246X.1996.tb01546.x.

Wu, P., and H. S. Hasegawa (1996b), Induced stresses and fault potential in Eastern Canada due to a disc load: A preliminary analysis, Geophys. J. Int., 125, 415–430, doi:10.1111/j.1365-246X.1996.tb00008.x.
* * *

---

## Author Comment (AC1) · 28 Mar 2018

I thank the reviewers for their thorough and helpful feedback. As noted by one reviewer, it is a difficult task to get the balance of detail right between different sections of a review paper, but the suggestions for additional material (and references) made by both reviewers will enhance the existing text. I will revise the article in light of their comments, which I think can be done in a way that does not make the final version too cumbersome.

My aim in writing this article was to focus on the historical, cutting edge, and future perspectives of research into Glacial Isostatic Adjustment (GIA). This necessitates a

brief description of the technical aspects of GIA, but my preference was not to delve into the theoretical details as this is covered extensively elsewhere. I will check carefully that appropriate references are included in this respect, and I thank the reviewers for picking up on technical issues and terminology that are currently not clear. I will also look again at the organization of the material in section 2.2 to ensure a more logical flow.

The reviewers have brought to my attention a number of additional historical references that I will be delighted to include, and they also provide useful insight into fields in which I am less familiar. Both reviewers note that the section on future perspectives may reflect my personal view on the future direction of GIA. However, as one reviewer notes, this section does indeed reflect discussions I have had with scientists from a wide range of disciplines as well as input from the wider community during workshop discussions, and this will be acknowledged. This section of the article is intended to serve as a catalyst for new research, and my hope is that it will be read by researchers from outside the traditional GIA community, who will take the relatively simple concept of GIA and apply it in novel ways in their own fields.

Pippa Whitehouse, 28/03/18

---

## Author Response (AR1)

**Response to Reviewers' Comments**

I thank the reviewers for their constructive feedback and address each of their comments in turn below. My responses are in blue. Where changes have been made to the manuscript in response to a reviewer's comment I have not quoted the revised text within this rebuttal document, but refer the reader to a relevant line number in the 'track changes' version of the revised manuscript attached below. All line numbers quoted in my responses reflect line numbers in the 'track changes' version of the revised manuscript.

**Reviewer 1 (anonymous)**

**1. General comments.**

This is a comprehensive review of the Glacial Isostatic Adjustment (GIA) problem, in which the emphasis is about the historic development, the recent advances and possible future directions. Despite the word 'modelling' appears in the title, there is little about the technical aspects of GIA and as a consequence the approach is rather qualitative and focussed on the description of the geophysical processes involved in GIA. A series of nice illustrations are proposed, useful to understand how GIA operates on global and regional scales, something that is certainly of interest for beginners in this field. The strength of the paper is in its completeness and organisation, which make it very easy to follow despite its considerable length. The weakness is in some missing details and references and sometimes in a too involved description of the various topics. In some parts, the paper seems to have been written in a rush, so that some smoothing and re-thinking is recommended before it can be considered for publication.

Thank you for the positive feedback. As you mention, my focus in writing this article was on the historic development, recent advances, and future directions of GIA. This necessitates a brief description of the processes associated GIA, but the reader is referred to appropriate references to uncover the technical details of GIA modelling (lines 47-49, lines 194-195). Following the guidance of the reviewers, I have attempted to balance the level of detail across all sections and ensure that the whole article reads smoothly.

I have a series of minor to moderate remarks, various considerations, notes and hints, listed in the following, which I hope are useful to improve an already very good contribution.

**2. Specific comments**

Line 1ff. In some previous works (e.g., DOI:10.1007/s10712-016-9379-x), a distinction is made between the Earth's response to past ice melting and that to present ice melting. Of course the physical principles are the same, but a few explicatory words can be of help. For example, when dealing with low-viscosity regions in Section 4.3, it should be clear that in this case we are dealing with the GIA caused by recent melting.

The distinction between past and present melting is important for low viscosity zones and an edit has been made to this effect in Section 2.5.1 (lines 563-564) where low viscosity zones are first introduced. It has not been included in the abstract to keep it simple and succinct.

L6. Not only to 'global' ice sheets; in some parts GIA is in response to regional ice sheets (e.g., again Section 4.3).

All ice sheets are regional, but the current wording reflects the fact that changes to all ice sheets must be considered when calculating relative sea-level change. Text unchanged.

L7. Actually, GIA 'follows' the entire history of glaciations, which is 100s of kyrs long. I do not think GIA should only be viewed only as the response to glacial unloading. Perhaps some words to distinguish between 'post-glacial rebound' and GIA can help, in this respect.

The purpose of the text on lines 8-9 is to highlight the fact that GIA is a relatively rapid process when considering geological timescales. The opening sentence of the abstract (lines 5-6) clearly states that GIA reflects the response to both glaciation and deglaciation. A definition of 'postglacial rebound' is now included on lines 6-8.

**L18. (LGM, ~21,000 years ago).**

Text added.

L44. GIA can be "seen" (and it was actually seen) by looking at the migration of the shorelines in northern Europe, as summarised here. However, in the whole paper there is not so much attention about the general effects of GIA on the time evolution of the past Earth's topography (e.g. former "land bridges" are only mentioned at L759). May be it could be useful to look at the literature of contemporary GIA investigators, in this respect. I am sure that a little section about this important aspect of GIA modelling would add some value to this review.

In the interests of not increasing the length of this article further I have not included a separate section on the impacts of topographic change or shoreline evolution, although their influence on a range of processes is discussed in sections 3.1 (Ice dynamic feedbacks), 4.2.2 (Coupled GIA-ocean/atmosphere modelling), and 4.2.3 (Coupled GIA-surface process modelling). The role of GIA in defining past migration routes is now mentioned in section 1.1 (line 58).

L52ff. I suggest to change item (viii) into "interpreting the gravity field of the Earth and its rotational state" or something like that.

**Text added on line 57.**

L60ff. This is a remarkable historical overview, indeed. I have some miscellaneous observations. *i*) Did Jamieson only consider elastic deformation? *ii*) Was his work influenced by the ideas of Airy about isostasy? *iii*) It is probably important to mention, for later comparisons, the famous Haskell value for mantle viscosity. *iv*) Regarding uniform viscosity Earth models, the Darwin spherical model has had a role in the development of GIA models (see the review of Peltier, 1974, who should be quoted e.g., at L176 between O'Connell and Chatles, in my opinion). *v*) I am surprised to see no mention to the fundamental work of Love, who with his "Love numbers" certainly prepared the advent of the modern GIA models, starting from Farrell and Clark (1976).

- i) Jamieson (1865) did not use the terms elastic or viscous, but he does mention that it is not clear whether the deformation of the land was 'gradual or sudden'. There is insufficient information in the original article to go into more detail so the text of this manuscript is unchanged.
- ii) I can find no evidence of Jamieson having been aware of Airy's work on isostasy.
- iii) The Haskell (1935) value for upper mantle viscosity is now mentioned on line 169.
- iv) A reference to Peltier (1974) has been added on line 170; the reader is encouraged to consult the referenced articles if they require further information on the early development of viscous Earth models.
- v) A reference to the work of Love (1919) is now included on line 337.

L175. I think 'is the equipotential' is better than 'is an equipotential'.

Text edited on line 179.

L176. This can be shortened into 'in the absence of winds and currents'.

The current wording was chosen to reflect use of the phrase 'dynamic ocean topography' in the wider literature when discussing departures of the sea surface from the equipotential surface. Text unchanged.

L180ff. This is nice illustration, in words, of the concept behind the 'k' Love number.

Thank you.

L187. I would suggest 'constrained' in lieu of 'determined'.

Original text preferred; no changes made.

L190ff. The role of the sea level equation (SLE) in GIA modelling has been recently reviewed in detail in DOI:10.1007/s10712-016-9379-x.

Reference to the review by Spada (2017) is added on line 195, as well as earlier on line 49.

L190ff. Eq. (1) and the material that follows is fairly good, but it can be improved, I have a suggestion. Why not starting with the SLE in the native form S=N-U (new equation 1)? This would be helpful, since in this way one

can define relative sea level change (S) and absolute sea level change (N), and vertical displacement (U) since the onset. These are quantities of fundamental importance for a full understanding of the remainder of the paper. If this is agreed, Eq. (1) becomes Eq. (2) and so forth... Also, the unnecessarily cumbersome symbol DeltaSL could simply become S and so on al L196 and L201 (and in other places, I think).

This is an excellent suggestion. An additional equation has been included defining the terms mentioned above and all other equations have been re-numbered.

L194. Given that the unknown in the sea-level equation is S (provided that the history of the changes in ice thickness I is assumed to be known), one may wonder how surface displacements U or horizontal displacements V are obtained for e.g. comparison with GNSS data, discussed later in the paper.

Edits on lines 201-203 now explain that sea-level change is determined by calculating changes to the shape and height of the sea surface and displacement of the solid Earth.

L196: ... is the change in \*relative\* sea level...

Text edited on line 207.

L201. This can be rearranged into: ... are convolutions in time (over the ice sheets history) and in space (over the surface of the ice sheets and of the oceans....).

The original phrasing is retained to account for the fact that ice and ocean load changes both vary over both space and time.

L204. It is not actually a uniform shift 'in the geoid', it is a uniform shift in *relative* sea level, according to the equation labelled by (1) in the present form of the paper.

**Text edited on line 215.**

L209. The definition of eustatic sea level is important in GIA modelling, but here it is only mentioned *en passant*. Seen the beautiful historical Section 2.1, some words could be certainly spent on the work of Edward Suess on the concept of eustasy in his book "*La Face de la Terre*". To be completely fair (and admittedly pedantic), eustasy is a static concept, only dependent on the amount of melt water change, on the density of water and on the area of the oceans surface. However, in his Eq. (2), the author has accounted for variations of the area of the oceans by a time-dependent A\_o(t). This is a dynamic effect, however, since it also depends e.g. on Earth's rheology. So, I suggest to warn the reader about this caveat, or to re-write the SLE using an effectively constant A\_o. I would personally prefer the first solution. I also note that in the recent literature the term 'eustatic' has been substituted by '*barystatic*' following the work of Gregory et al. (2013), something that could be mentioned or not. I finally observe that when a time-dependent A\_o(t) is accounted for (this is the case for migrating shorelines), the sea level equation becomes non-linear; otherwise it is linear for an effectively constant A\_o. This an important qualitative difference that should be mentioned, in my opinion.

I thank the reviewer for making me aware of the extensive work of Eduard Suess, which makes for very interesting reading! I have now included mention of the fact that Suess was the first to use the term 'eustasy' on lines 222-223, but since his explanations for local and global sea-level change do not consider the role of the global ice sheets I have not included further discussion of his ideas in the main text. In this paragraph (lines 220-233) a caveat relating to the time-dependence of ocean area is included, along with mention that this makes the sea-level equation non-linear, and a definition of the term 'barystatic'. Reviewer 2 also requested that the term 'eustatic' be discussed in a little more detail so some of the edits in this paragraph are in response to their suggestions.

L212. They are not really 'perturbations'; maybe they are 'terms'.

**Text edited on line 232.**

L222. The Maxwell rheology is describing a fluid behaviour, not a solid behaviour. In fact, in a creep experiment, just immediately after the instantaneous elastic response, the dashpot of the Maxwell body works as a Newtonian viscous fluid. The same at L223, and in other places like L570, for instance.

In all instances (lines 335, 336, 666) the text is changed to refer to a 'Maxwell body' since there is some discrepancy in the literature over the use of the terms 'Maxwell solid' and 'Maxwell fluid'.

L226. If a power law is assumed, the Love numbers formalism is not viable since the problem becomes non-linear.

**Text edited on lines 339-340.**

L234. The lithosphere is far from being purely elastic (see e.g. Ranalli, *The Rheology of the Earth*, or DOI: 10.1029/RG021i006p01458, or the papers by Burov, e.g., DOI:10.1029/94JB02770). The main point is "why the elastic lithosphere approximation is so common in GIA studies"? (despite the evidence for a complex rheological profile).

Text has been added on lines 348-349 to reflect the fact that many models define the lithosphere as a high viscosity layer.

L250. What 'self consistent' means here?

**Text deleted.**

L247ff. All these extensions of the sea level equation make it a non-linear equation; this is an important point. See also my comment to L209 above.

Mention that the sea-level equation is a non-linear equation is now included on line 230.

L256ff. It can be worth to say that the effects of Earth rotation introduce a very long wavelength pattern mostly characterised by harmonic degree l=2 and order m= +/-1 terms. This high-energy component of sea-level change is clearly visible in the form of large lobes in the maps GIA fingerprints, see e.g, http://dx.doi.org/10.1016/j.gloplacha.2016.05.006.

Text on line 282 has been edited to include mention of the fact that rotational feedback results in long wavelength perturbations.

L256ff. I would rephrase as follows "Since GIA alters the mass distribution of the Earth, it changes its (offdiagonal) moments of inertia, which in turn perturbs..." or similar.

In order to keep the language simple, and to avoid having to define additional terms, the original text has not been edited.

L261. Why 'over longer time scale'? Actually, rotationally induced Earth deformations also occur on short time scales, even elastically (i.e., instantaneously).

Text on lines 280-281 edited to reflect the difference between elastic and viscous deformation.

L262. Again, I do not capture the rationale for separating short from long time scales.

This is now more clearly explained in the text on line 280-281.

L274. I would avoid the term 'transient', since this term can also refer to non steady-state (e.g., Burgers) rheological effects. I think that 'time-dependent' could be a possible alternative.

**Text edited on line 294.**

L285. I am not sure that the meltwater fingerprints can be immediately associated to (or 'based on') the theory of Woodward. Indeed, Woodward used a rigid Earth and ignored the oceans self-attraction (only accounting for the gravitational attraction between the point ice load and the ocean mass). Hence, his fingerprints are approximations of the 'modern ones' based on a more realistic modelling.

Text on line 305 edited to state that the idea of fingerprints 'builds on' the theory developed by Woodward.

L285 and L289. The term 'fingerprint' in the GIA context has been coined by Plag and Jüettner (2001) and adopted in numerous studies since then (see Plag HP, Juettner HU Inversion of global tide gauge data for present-day ice load changes - scientific paper – Mem Natl Inst Polar Res 54:301 special issue, 2001), and this should be fully acknowledged.

Thank you, I was not aware of this. A reference to the article by Plag and Jüttner (2001) is added on line 309.

**L293ff. One may simply wonder \*why\* peripheral bulges regions exist!**

A description of the processes that lead to the formation of peripheral bulges is included in section 2.1 (lines 157-158).

L297. If my hints at L190ff above are followed, the reader will be greatly facilitated here, where the concept of absolute sea level is utilised. Why 'mean'? I do not think it is necessary.

The term 'mean' is deleted on line 317.

L300 and 301. I am afraid I do not understand this couple of lines. Global (absolute) sea level change obtained from altimetry is normally corrected for the effects of GIA, so ocean syphoning and all the other processes involved are certainly taken into account. I think the paper of Tamisiea (2011) is dealing with other aspects of GIA. It is possible, however, that I am missing the point, here.

This sentence (now on lines 320-321) simply states that ocean syphoning, in response to the subsidence of peripheral bulge regions, must be taken into account when interpreting altimetric measurements of absolute sea level. It is my understanding that this is the process described in Tamisiea (2011). Text unchanged.

L305. The 'continental levering' effect should be mostly visible far from the polar regions, right?

Continental levering will also be important in polar regions. For example, the Bering Strait and the eastern part of the Siberian margin (assumed unglaciated during the LGM) are likely to have been exposed during the LGM lowstand. Levering will have come into play as these regions were flooded during deglaciation. Text unchanged.

L312ff. Perhaps, it can be of interest to note that before the advent of the so-called pseudospectral method, a fully spectral approach was utilised (see e.g., the paper by Plag and Juettner quoted above). Regarding the methods in general, it could be instructive to explain \*why\* certain methods are used instead of others. For instance, the pseudo-spectral method is now standard when dealing with spherically symmetric Earth models with linear rheology. Similarly, the finite-element approach of Wu and co-workers is motivated by the introduction of a power-law rheology (for which the superposition principle does not hold), and so on...

Text in this paragraph (lines 362-372) has been edited to include mention of the motivation for using different methods to solve the sea-level equation.

L320. Here and in other places, the word 'self-consistent' should be used in a more specific way. I think that in the context of GIA the term 'gravitationally self-consistent' has been introduced to say e.g. that the change in the shape of the oceans determined by solving the sea level equation is consistent with the gravitational field (the oceans surface is an equipotential ad mass conservation is ensured). Similarly, 'topographically self consistent' indicates that the solution of the sea level equation for a variable topography is consistent with the present day topography (and with the gravity field). See the works of WR Peltier, where (I think) this terminology has been introduced first. See also point L250.

The text has been edited to explicitly refer to a 'gravitationally self-consistent solution' on line 371. Where the phrase is originally used (line 28) care has been taken to explain that it refers to the fact that the shape of the ocean surface is defined by the gravitational field. In other instances the phrase 'gravitationally self-consistent' is used; the phrase 'topographically self-consistent' is not used anywhere in the manuscript.

L321. I do not think that \*all\* integral equations need to be solved by iteration. Iteration is often invoked to solve non-homogeneous Fredholm equation of the second kind, which is the type of equation the sea level equation is. In any case, most importantly, the physical reason for which iterations are needed is that the change on the ocean mass distribution is not known a priori, contrary to the ice distribution.

Text on lines 371-372 has been edited to reflect the fact that an iterative approach is required to account for the fact that the time-dependent change in ocean loading is not known a priori.

L322. The Section on Data is very smooth. Regarding 2.2.3, I only observe that before GPS data, very long baseline radio-interferometry (VLBI) data have also been employed to test GIA models and to constrain the Earth's viscosity profile. This can be traced easily in the literature.

The use of VLBI (and other ground-based geodetic techniques) to study GIA is reviewed in King et al. (2010); a reference to this article is now included on line 460.

L400ff. I do not fully agree on the role of horizontal GPS observations; their recognised sensitivity to the shallow upper mantle rheology and to the presence of lateral variations in the Earth's mantle properties should be regarded as an advantage, not as a limitation.

Good point. Text on lines 476-477 changed to reflect the positive implications of horizontal deformation being dependent on lateral earth structure.

L402ff. Similarly, I do not think GIA model predictions are typically provided on a reference frame whose origin lies at the centre of mass of the solid Earth. Quite often, instead, they are given in the reference frame of the center of mass of the whole Earth system, so that they can be directly compared to geodetic information (actually, in GIA modelling the transformation between the two frames is straightforward). I agree on the reference frame origin uncertainties, which is indeed a major problem.

It is true that GIA model predictions need to be in a centre of mass reference frame (often referred to as 'CM') for comparison with geodetic observations, but the original calculations are usually carried out in a centre of solid Earth reference frame (often referred to as 'CE') – see Kierulf et al. (2014) for a detailed discussion of this point. Text unchanged.

L416ff. This is true, but I think the main problem with GRACE is that it observes gravity field variations due to all surface and internal sources (including e.g. mantle dynamics associated with non-GIA processes, post-seismic deformations, and so on). I also note that some very useful insight into the actual meaning of GRACE data has been recently given by BF Chao in DOI:10.1007/s00190-016-0912-y.

The text on lines 495-496 has been edited to re-iterate the fact that changes to the gravity field also reflect non-GIA-related processes.

L444ff. I am not really getting the point here. Can examples and/or citations be given to support or explain this sentence?

The opening sentence of this paragraph (line 521-522) has been deleted.

L454. As far as I know, a possible interaction between GIA and seismicity has been first proposed by Gutenberg and Richter in their book *Seismicity of the Earth and Associated Phenomena* (1949), and there is a long story of relevant contributions to this field since the 90s, probably listed in the bibliography of Dr. Steffen's works. Some considerations along these lines could help to put flesh on the bones of this succinct subsection.

The link between GIA and seismicity is only briefly mentioned in *Seismicity of the Earth and Associated Phenomena* (1949, p101) so this reference has not been included, but following the advice of Reviewer 2 a number of other references on this subject have been included in this section and in the section that describes the approaches used to model GIA (section 2.3.2).

L467. I agree that the definition of intervals for the viscosity of the upper and lower mantle has been (and is) a key-result in GIA modelling. But I also think that the recognition that GIA data requires a rheologically layered mantle (i.e. a viscosity jump between the upper and a the lower mantle) is even more important. It would be very useful to trace this key result in the literature.

A sentence has been added on lines 558-560, and a number of references included, providing insight into this result.

L475ff. Although the physical process is the same, it is important to let the readers know that contrary to the studies mentioned at ~L470, in those quoted here the source of GIA is recent ice melting from small size glaciers and ice caps. By the way, I definitively agree that defining these low-viscosity layers is a key result of GIA modelling.

**This point is now clarified in the text on line 563.**

L479-L491. This paragraph is OK, but still it is difficult to see sharp key results. Itemisation can help? May be the text can be modified accordingly. In any case, since the 70s the Toronto University GIA school has given important contributions into the definition of the ice volume change since the LGM with the help of GIA modelling, and this should be fully recognised, in my opinion.

The text in this paragraph (lines 569-587) has been re-organised to highlight the key results. A reference to Peltier (2004) has been included on line 572.

L492ff. I realise that summarising all the attempts to constrain the configuration of individual ice sheets in a short paragraph is not easy. But here one is left with the impression that all the studies mentioned converge to similar results. However this is not the case; for instance, as the author of this manuscript knows, different studies show diverging results for the history of melting chronology of the Antarctic ice sheet during the late Holocene. Actually, resolving this uncertainty could constitute one of the future challenges in GIA modelling for Section 4 below.

Text is added on lines 594-595 to reflect the fact that there is still considerable uncertainty regarding the evolution of the global ice sheets during the last glacial cycle. This point is also now reflected in the Conclusions (lines 938-940).

L501. The sentence should be probably tuned differently for the Greenland and Antarctic Ice Sheets. In any case, GIA corrections are particularly uncertain because of the uncertainties about the melting history of Antarctica.

Text has been added on line 603, noting that the contemporary GIA signal across Antarctica is still very uncertain.

L522ff. Just a comment. There are uncertainties associated with limited data availability and modelling capability. However, I think that the uncertainties described by Tamisiea (2011) are of different nature, resulting from genuine misunderstanding of the physical meaning of the various terms of the sea-level equation.

The article by Tamisiea (2011) does indeed discuss issues associated with the inconsistent use of terminology, but the reason for referencing this article here is because it attempts to quantify the uncertainty associated with the contemporary GIA signal. Text unchanged.

L584. Yes, this is an important question. Concerning global GIA models, I do not think we are now in the position of saying that 3D non-Newtonian models perform better that 1D Maxwell models in explaining e.g. the sea-level variations observed during the Holocene. So, I do not know whether an increased model complexity is indeed required.

The effect of accounting for lateral Earth structure in a GIA model is quantified and compared with the precision of different data types (lines 680-684). It is left to the reader to decide whether consideration of lateral structure is necessary, depending on their area of interest. Text unchanged.

L615. I am not really an expert in sedimentary isostasy, but I know that sometimes sediment loading and sediment compaction occur at a very small scale. In view of that, is there some indication of the spatial resolution of the GIA models in which these effects are taken into account? Maybe some information could be conveyed to the readers in this respect. I am also curious about how mass is conserved in these sedimentary GIA models, since this would require detailed information about the sediments (re)distribution, etc...

A sentence has been added on lines 736-738 highlighting the fact that sediment redistribution and compaction can indeed take place over small spatial scales, and that the modelling approaches described in this section are primarily suited to studying the large-scale effects of sediment redistribution. The approach used to conserve the

mass of the sediment is discussed in a number of the articles referenced in the manuscript, e.g. Ferrier et al. (2015; 2017); Kuchar et al. (2017).

L653. The total mass change should be zero, by mass conservation; probably 'mass redistribution' is meant here? A few lines below (L669): I am missing why an inverse approach is neither dependent on the ice loading history nor from the Earth structure. What does it depend upon? Overall, this whole Section 3.4 is definitively not written for newbies and is somewhat confusing, in my opinion.

Text is altered to read 'spatial pattern of mass change' on line 751. The inverse approach discussed in this section refers to a data inversion: the contemporary GIA signal can be isolated by combining different data sets because they each have a different sensitivity to GIA. Text on lines 748-750 and 7768 has been altered to clarify this point.

L676. Section 4: future directions. This is a very nice (and very personal) view of the possible future developments of GIA modelling. I do not have specific comments on that.

Additional text has been included in the Acknowledgements (lines 968-971) to reflect the input of the wider research community in contributing to the ideas discussed in this section.

L827. A very short conclusion that sounds a bit vague. I would have preferred a few (better if itemised) statements summarising three or four take home messages.

The Conclusions have been rewritten so that they summarise key results and a number of important areas of future research.

Hope this helps
* * *
Review papers can be a dangerous task to undertake, and can also be challenging to comment on. Arguments of different referees can be condratictory: too long, too short, too many references, too few, not all is covered, too much is covered, too much focus on one issue, chapters order should be the other way around and so on and so forth. As an author, it is challenging to satisfy all those views and partly needs a thick skin, also once it is published and more people comment on it.

From my perspective, Pippa Whitehouse reviews glacial isostatic adjustment (GIA) modelling very well and I think that this paper will be highly regarded and cited, especially as it draws a bow from the historical development over recent advances to possible future directions posing a few interesting research questions to take on for the wider community.

The paper is in most parts well written and additionally supported by a couple of figures that help explain major GIA-related processes. Much thought was given on recent and future developments and may likely reflect many personal views, I note though many discussions that Pippa does on conferences, thus I think she has a good overview of ongoing and future works.

I recall the GIA workshop in Reykjavik in September 2017 where the 10 top research questions in GIA research were collected. I guess some of them also arise while reading this paper and perhaps it is an option to pick that up in the Conclusions pointing to the SCAR-SERCE website.

There also are a few paragraphs/sections that are very brief with a few common references while others are well developed with a large number of references. I guess this is due to the personal research interest of the author, and I try to give some help below so that brief parts can be extended.

I hope that my comments in the following are taken in the positive manner with which they are intended. I apologize right away that some comments and mentioned references are due to my "Fennoscandia-biased" eye, so please take them as suggestion only, especially when I write "I'd suggest"...

Thank you for your positive comments. Responses to specific queries are addressed below. In the interests of not making this manuscript (or reference list) too much longer, I have taken the opportunity to regard a couple of your comments as 'suggestions only'; I hope this is acceptable. The suggestion to draw on the list of research questions proposed by the research community during the IAG/SCAR-SERCE GIA Workshop held in September 2017 has been taken up – see edits to the Conclusions and Acknowledgements.

**General remarks**

L12: I would not agree with "in detail" when you mean "field of GIA" - there is so much more to discuss - but I would support it when you say "field of GIA modelling".

The phrase 'in detail' has been deleted on line 13.

First paragraph of Introduction: I'd suggest to add 1-2 sentences about the term "postglacial rebound" which you mention three times in this paper, e.g. L780 "postglacial rebound is not the only GIA-related process". It would be good for the reader to learn about this term and how it relates to GIA, especially as it is still often used synonymously in publications.

A definition of 'postglacial rebound' is now included on lines 6-8.

L44-59: as your paper has next to modelling a strong bias to sea level, it is perhaps good to mention a couple of other review papers the reader can look at, e.g. Whitehouse (2009, I really missed that in the references), Steffen & Wu (2011), Milne (2015), Spada (2017)

A reference to these articles is included on lines 48-49.

L64/65: The sentence is per se ok but I note that Celsius was not the first to cut marks for the sea level in rocks, see Mörner (1979). Celsius' intention though was to allow rigorous measurements, especially for future generations. See Martin Ekman's new book (2016) about Celsius, section 5.4 therein.

This sentence has been rephrased (lines 67-69) to reflect the fact that Celsius was not the first to cut sea-level marks.

Section 2.2: This section discusses only GIA models applying the sea-level equation but misses among others flat Finite Element models. Also GIA-fault models are not yet connected to it. I'd suggest to either alter the title slightly including the SLE or add another short section with a discussion of models not involving the SLE, see my comments on L216-218.

The title of section 2.2 has been edited to reflect the fact that this section only considers models that solve the sea-level equation. The other types of models mentioned by the reviewer are now discussed in section 2.3.2. See also response to comments associated with L216-218.

L183/184: you can well mention here that the "system Earth" has further contribution that you will pick up later.

It was decided not to include discussion of non-GIA processes at this point. No edits made.

L209: I'd suggest to add references Lambeck et al. (2001) and Milne et al. (2002) here and add a few words on the issue briefly discussed in both papers, that is that inconsistent use of this term has caused some confusion earlier, thus one should be aware that some publications may use it in a wrong way.

References to both papers have been included on lines 224-226, along with a brief discussion of the pitfalls associated with the term 'eustatic'. Additional text in this paragraph reflects responses to similar comments by Reviewer 1.

L216-218: Although this paragraph intends to make a link to following sections, I became irritated while reading the next sections. I expected a discussion of the different quantities of the SLE, especially as you began in 2.2.2 with the solid Earth, thus I expected the ice thickness evolution next and so on. Putting 2.2.2 aside (and perhaps move it somewhere else) Section 2.2 would deal with the SLE only because 2.2.3-2.2.5 deal with SLE but not with GIA models as indicated in the title of 2.2. So a suggestion could be to rename section 2.2 to "Development of the sea-level equation" or something similar, section 2.2.1 to "The original form of the sea-level equation", moving 2.2.2 after 2.2.4 to have old 2.2.1, 2.2.3 and 2.2.4 as new 2.2.1, 2.2.2 and 2.2.3. Then put old 2.2.2 together with 2.2.5 and perhaps other GIA models without application of the SLE including GIA-faults models (R. Steffen et al. 2014a) to a longer new "Numerical methods for modelling GIA" section with old 2.2.2, missing GIA without SLE, old 2.2.5 to new 2.3.1, 2.3.2 and 2.3.3. This should also fit the title of section 2.

This section of the article has been re-ordered, following the suggestions of the reviewer. Section 2.2 is now titled 'Development of the sea-level equation', and contains sections on 'The original form of the sea-level equation' (2.2.1), 'Extensions to the sea-level equation' (2.2.2), and 'Confirmation of early theories and implications for the interpretation of sea-level records' (2.2.3). Section 2.3 has been added; it is titled 'Numerical methods used to model GIA', and contains sections on 'Representation of the solid Earth' (2.3.1) and 'Modelling approaches' (2.3.2). The majority of the text in sections 2.2 and 2.3 is unchanged (except in response to other reviewer comments), but additional text has been included in section 2.3.2 on modelling approaches that do not solve the sea-level equation and the modelling of glacially-induced faults.

Section 2.2.5: I'd suggest if possible add a few words on the temporal and spatial resolution of the models and how it has changed over the years. Anything easy to say about the run time for such models - really general in terms of minutes, hours or days?

Given the additional text already added in response to other suggestions by the reviewers in what is now section 2.3.2, it was decided not to include this additional information in the manuscript. No edits made.

Section 2.3: (1.) You briefly mention optimal location later but I'd suggest to place a few words already here. I note that there is not only the study by Wu et al. (2010) on optimal GPS, later on Steffen et al. (2012, 2014) also discussed optimal gravity and RSL data, respectively. (2.) You miss a quite interesting, though also difficult to analyse data set: tide gauges - just mentioned two times in the manuscript. However, it is a crucial data set in all sea-level rise discussions. I'd suggest to add a section here.

(1) A sentence has been added on lines 399-400 supporting the collection of new data from locations that are optimally sensitive to the details of ice history and Earth rheology, and the three suggested references are included. (2) A couple of sentences on tide gauges are included on lines 424-427.

Section 2.3.3: I miss the studies by Nerem & Mitchum (2002) and Kuo et al. (2004) using SAR together with tide gauges.

The suggested references have been added on line 462.

L396-398: I'd suggest to mention the GIA-frame approach introduced in Kierulf et al. (2014) that deals with this problem and can be regarded quite successful when checking the discussion therein.

This additional information has not been included in the manuscript. No edits made.

L398-401: Getting information on the subsurface structure from horizontal velocities was the aim of Milne et al. (2004), Steffen et al. (2007) and Steffen and Wu (2014). The latter two papers are quite technical and I admit rather complex for reading but they support your conclusion that horizontal rates have not yet been used to their full potential. However, studies exist.

Text has been edited in response to Reviewer 1, and a reference to Steffen and Wu (2014) has been added on line 477.

L416: suggest to add "e.g." to the references

Changes implemented on line 493.

L424-426: in view of the relationship between uplift and gravity change, the study by Olsson et al. (2015) should be mentioned.

Reference added on line 503.

L439: strongly suggest to change "may not be" to "are not"

Text edited on line 516.

L449/450: Argus et al. (2014) should be mentioned along the three papers.

Reference added on line 527.

Section 2.3.6: one of the rather short sections where much research has been undertaken, see the works by Wu, Johnston, Klemann, Kaufmann, Lund, R. Steffen etc. This section should be expanded. Stewart et al. (2000) is certainly a key paper but important studies were already published in the late 70s and much research was triggered by the studies of Spada et al. (1991), Wu & Hasegawa (1996a,b) and Arvidsson (1996), to name a few.

This section (Sect. 2.4.6) has been expanded, although since it comes under the broad heading of 'Data' (Sect. 2.4) some purely-modelling results have not been included. Additional references have been added, both in this section and in the section on 'Modelling approaches' (section 2.3.2).

L455/456. The unloading is not only able to trigger postglacial faulting but also more recent (historic) earthquakes can be linked to GIA, see Brandes et al. (2015). Since a few years the term "postglacial fault" or "postglacial faulting" is thus under discussion, see e.g. Lund (2015). My personal preference is "glacially induced fault".

The phrases 'glacially-induced faulting' and 'glacially-induced earthquakes' have been adopted in Section 2.4.6.

L458: triggering slip on pre-existing faults was a major result of R. Steffen et al. (2014b).

This reference is now included on line 535.

L495: I'd suggest to add Nordman et al. (2015) to Fennoscandia (Lev Tarasov's North-European model), and Lambeck (1995) for the British ice sheet. What about Patagonia and the work of Ivins and James (2004)?

The article by Nordman et al. (2015) has not been included since it contains very little information about the ice models used in that study to drive the GIA modelling, and as far as I can tell they were not produced by 'comparing GIA model output with a range of data sets' (line 589). Patagonia is not mentioned in this paragraph because the ice that grows there is not large enough to be defined as an 'ice sheet' (although the article mentioned by the reviewer is cited elsewhere in the manuscript). Reference to the article by Lambeck (1995) has been added on line 591.

L498: Lambeck et al. (e.g. Lambeck et al. 2014) and Tarasov also work on global models and should be named.

Reference to Lambeck et al. (2014) added on line 595; I'm afraid I could not find a suitable reference for the Tarasov global model so this is not mentioned in the text.

L577: Klemann et al. (2008) should be cited for the effect of plate boundaries.

Reference added to lines 673-674.

L578: Wang & Wu (2006a,b) also analyzed the effect of a 3D lithosphere (it's in the paper titles).

References added on line 675.

Section 3.3: In view of this section, L677/678, Section 4.2 and the Conclusions you can state that GIA is part of Earth System Modelling and should always be investigated in an interdisciplinary context.

A statement about the importance of including GIA in future Earth System modelling efforts is now included on lines 776-777.

L688-691: I miss a reference here.

I am currently involved in a study that seeks to quantify the magnitude of this component of post-glacial sealevel rise; to my knowledge it has not previously been quantified and therefore I am unable to provide a reference.

L725: This relates to the concept of "underwater GPS". Future measurements may help solving this question and it is perhaps worth to mention that research is going on in this field (e.g. Ramesh et al. 2016, Honsho & Kido, 2017).

A reference to Honsho and Kido (2017) has been added on line 825.

L729: I'd suggest stating "use of satellite gravity data". Terrestrial gravity data are not a gap-filler for GPS in view of the effort to perform a single measurement.

Text altered on line 829.

L777: The studies by Schmidt et al. (2013) and Kutterolf et al. (2013) should also be mentioned.

A reference to Schmidt et al. (2013) is now included on line 877, and a reference to Kutterolf et al. (2013) is inserted later in the paragraph in a more relevant location (line 882-883).

L830: "central component of any GIA model" implies that flat FE models are no GIA models..., suggest "any GIA model"="the majority of GIA models"

This part of the sentence has been deleted.

Figure 1: In view of the basic equations the outputs are not complete. Rotation and stress are missing, which are both quantities that you discuss in your manuscript. Should be added.

Labels relating to rotation and stress have been added to Figure 1.

**Technical corrections** L97: The first ice(-)cap appearance should be without dash Thanks, this has been corrected.

[revised manuscript text omitted]